**RESEARCH**

# Metapangenomics of the oral microbiome provides insights into habitat adaptation and cultivar diversity

Daniel R. Utter[1*] , Gary G. Borisy[2], A. Murat Eren[3,4], Colleen M. Cavanaugh[1*] and Jessica L. Mark Welch[3*]

* Correspondence: dutter@g. harvard.edu; cavanaug@fas.harvard. edu; jmarkwelch@mbl.edu
[1]Department of Organismic and Evolutionary Biology, Harvard University, Cambridge, MA 02138, USA
[3]The Josephine Bay Paul Center for Comparative Molecular Biology and Evolution, Marine Biological Laboratory, Woods Hole, MA 02543, USA
Full list of author information is available at the end of the article

## Abstract

**Background:** The increasing availability of microbial genomes and environmental shotgun metagenomes provides unprecedented access to the genomic differences within related bacteria. The human oral microbiome with its diverse habitats and abundant, relatively well-characterized microbial inhabitants presents an opportunity to investigate bacterial population structures at an ecosystem scale.

**Results:** Here, we employ a metapangenomic approach that combines public genomes with Human Microbiome Project (HMP) metagenomes to study the diversity of microbial residents of three oral habitats: tongue dorsum, buccal mucosa, and supragingival plaque. For two exemplar taxa, *Haemophilus parainfluenzae* and the genus *Rothia*, metapangenomes reveal distinct genomic groups based on shared genome content. *H. parainfluenzae* genomes separate into three distinct subgroups with differential abundance between oral habitats. Functional enrichment analyses identify an operon encoding oxaloacetate decarboxylase as diagnostic for the tongue-abundant subgroup. For the genus *Rothia*, grouping by shared genome content recapitulates species-level taxonomy and habitat preferences. However, while most *R. mucilaginosa* are restricted to the tongue as expected, two genomes represent a cryptic population of *R. mucilaginosa* in many buccal mucosa samples. For both *H. parainfluenzae* and the genus *Rothia*, we identify not only limitations in the ability of cultivated organisms to represent populations in their native environment, but also specifically which cultivar gene sequences are absent or ubiquitous.

**Conclusions:** Our findings provide insights into population structure and biogeography in the mouth and form specific hypotheses about habitat adaptation. These results illustrate the power of combining metagenomes and pangenomes to investigate the ecology and evolution of bacteria across analytical scales.

**Keywords:** Oral microbiome, Metagenomes, Pangenomes, *Rothia*, *Haemophilus parainfluenzae*, Population structure, Biogeography

## Introduction

The human microbiome encompasses tremendous microbial diversity. The growing recognition of this diversity and its importance for human well-being prompted a major effort to investigate the identity and distribution patterns of bacteria throughout the human body, the Human Microbiome Project [1]. More recent studies have focused on finer-scale patterns, such as the role of host individuality in determining microbiome composition, the number and diversity of strains that can co-exist within a habitat, and the distribution of strains across body sites [2–4]. However, the sheer numbers and genetic diversity of bacteria in even a simple real-world microbiome present significant challenges to study.

One approach to studying bacterial populations is metagenomics—the direct sequencing of the total DNA obtained from an environmental sample [5]. By circumventing the need for cultivation, metagenomics can afford deeper and more accurate insights into the genetic diversity of naturally occurring microbial populations [6]. The Human Microbiome Project (HMP) [1, 7] sequenced metagenomes from hundreds of samples from sites all over the human body. However, the use of metagenomic methods alone can be limited by the challenges inherent in associating short reads back to a single organism without combining sequences from separate strains [5, 8].

On the other hand, the genomic diversity and relatedness within a group of bacteria can be studied using pangenomes. The pangenome, the sum of all genes found across members of a given group, reveals the functional essence and diversity held within that group [9–11]. Pangenomics can identify core and accessory genes (genes shared and not shared by all, respectively) within a group of related bacteria, as well as the relationships between different bacteria based on shared gene content. Notably, relating genomes by gene content allows a phylogenetically naïve approach to compare genomes, so that any phylogenetic or ecological correlation that emerges from the comparison is informative [12, 13]. Because concepts of species pose challenges when working with bacteria, bacterial pangenomes may be generated at the genus or family level to illuminate gene sharing and the degree of relatedness within these larger groupings [14, 15]. However, the environmental distribution of groups and genes identified in the pangenome remain unidentified.

Combining pangenomes and metagenomes can offer broader perspectives into the adaptation of microbial populations to different habitats [16]. Pangenomes and metagenomes are complementary even when the organisms used for the pangenome were not isolated from the same samples whose metagenomes will be studied. A pangenome constructed from isolates collected at different times and around the world reveals the shared and variable gene content of the different organisms. The genomes can then be used for metagenomic read recruitment to investigate the distribution and biogeographical patterns of environmental populations and their genes [17–21]. Such approaches have also been applied to human epidemiology [22, 23] and the human microbiome [24–28]. Thus, by using a set of well-characterized genomes from members of a species or genus (i.e., a pangenome of cultivars) as a reference set to recruit reads from metagenomes spanning a variety of habitats, the relative frequency of each gene sequence in naturally occurring populations can be quantified. The ability of short-read mapping algorithms to map related but non-identical reads can be exploited to use reference genomes as reference points to probe the composition and structure of wild populations [29]. The combination of metagenomes with pangenomes, also

referred to as "metapangenomics" [13], reveals the population-level results of habitat-specific filtering of the pangenomic gene pool.

The oral microbiome is an ideal system in which to investigate microbial population structures in a complex landscape. Different surfaces in the mouth, such as the tongue dorsum, buccal mucosa, and teeth, constitute distinct habitats, each with a characteristic microbiome [30–33]. These microbiomes are dominated by a few dozen taxa with high abundance and prevalence [34–36], most of which have cultivable representatives from which genomes have been sequenced ([37], www.ehomd.org), making the system unusually tractable relative to other natural microbiomes. The microbiomes that assemble in the different oral habitats are clearly related to one another—composed of many of the same genera, for example—but are largely composed of different species [36]. For example, the major oral genera *Actinomyces*, *Fusobacterium*, *Neisseria*, *Veillonella*, and *Rothia* occur throughout the mouth, but their individual species show strongly differential habitat distributions. Individual species within these genera typically have 95–100% prevalence across individuals and make up several percent of the community at one oral site, but show lower prevalence and two orders of magnitude lower abundance at other oral sites [32, 34, 36, 38]. The reproducibility of taxon distribution across individuals, despite the frequent communication of the habitats with one another via salivary flow, suggests that founder effects and other stochastic processes are unlikely to explain the differences in species-level distribution and that these differences likely arise from selection. However, some apparent "habitat generalist" species, such as *Haemophilus parainfluenzae*, *Streptococcus mitis*, and *Porphyromonas pasteri*, can be found throughout the mouth [32, 34, 36, 38]. Altogether, the mouth is colonized by well-characterized bacteria that build distinctive communities in the different oral habitats in the absence of dispersal barriers. This setting presents an opportunity to investigate the genomic characteristics underlying the differential success of closely related species in different habitats.

Here, we combine pangenomes and metagenomes to investigate how genes are distributed across populations at distinct sites within the mouth. We focused on two exemplar oral taxa with high prevalence (> 95%) [31, 32, 34] and high abundance in the mouth: the species *Haemophilus parainfluenzae* and the genus *Rothia*. These two taxa represent the two oral biogeographic patterns, with *H. parainfluenzae* representing apparent habitat generalist species and the genus *Rothia* representing genera composed of habitat-specific species. Both *Rothia* spp. and *H. parainfluenzae* are sufficiently abundant—making up on average several percent of the microbiota at their sites of highest abundance—that metagenomic read recruitment to reference genomes can reliably sample their natural populations. As the basis for the analysis of natural populations, we constructed pangenomes using genome sequences from previously cultured isolates. We then investigated the degree to which each gene in the pangenome is represented in populations from the healthy human mouth using metagenomic data from the Human Microbiome Project [1]. We found that genomes can be clustered into distinct, nested genomic groups that show differences in abundance between habitats. Our results suggest a framework where bacteria are structured into multiple cryptic subpopulations, some of which match observed habitat boundaries.

## Results

### Metapangenome workflow and the environmental core/accessory designation

A metapangenome provides an integrated overview of how genes are distributed across reference genomes and across metagenomes [13, 29]. A conceptual schematic for how we used anvi'o [20] with the approach described by Delmont and Eren [13] to combine isolate genomes and oral metagenomes into a metapangenome is shown in Fig. 1 and Additional file 1: Fig. S1.

Construction of the pangenome rests on the definition of gene clusters, groups of genes that are close to one another in sequence space at the amino acid level (Fig. 1a parts 1 and 2). The presence or absence of gene clusters in individual genomes can be displayed so that sets of homologous genes are easily identified that are shared by all genomes, shared by subsets of genomes, or unique to a particular genome (Fig. 1a part 3). In parallel with the construction of the pangenome at the amino acid level, the distribution of each gene within the human mouth is assessed at the nucleotide level by mapping metagenomic reads against the entire collection of cultivar genomes (Fig. 1b part 1). The result of mapping is a value for "depth of coverage," hereafter simply "coverage," the number of metagenomic short reads that were mapped to a given nucleotide in a given genome; the coverage value serves as an estimate of the abundance

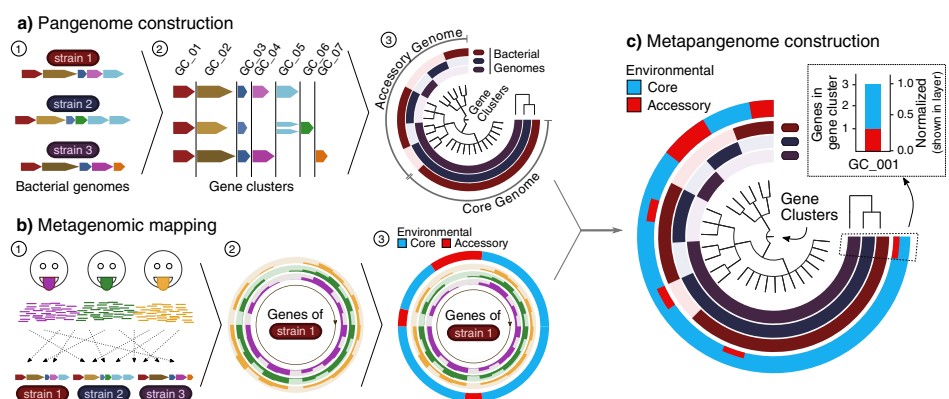

**Fig. 1** Metapangenomic workflow. **a** Pangenome construction. (1) All putative protein-coding gene sequences (colored block arrows) are extracted from each bacterial genome (colored bacilli above genes) to be included in the pangenome and (2) clustered into homologous gene clusters via blastp results grouped by the Markov Clustering Algorithm (sequence variants cartoonized as shades of the same color). (3) These gene clusters become the central dendrogram of the pangenome. Note that the gene clusters are organized by occurrence in genomes, not based on the order found in a particular genome. The detection of each gene cluster in a genome is visualized by filling in to indicate the presence or absence of each gene cluster across the genomes. The genomes are ordered by a dendrogram (top right) based on each genome's gene cluster content. **b** Metagenomic mapping. (1) The exact genomes as above in **a** are used as a reference onto which reads from metagenomes are mapped. Gene-level coverage for each gene is then calculated. (2) These coverages are plotted for all genes from a given genome to show that genome's gene-level representation in those samples. (3) Environmental representation is evaluated for each gene to decide whether that gene is environmentally core (gene's median coverage > 0.25 of the genome's median coverage across metagenomes from that environment) or environmentally accessory (gene's median coverage < 0.25 of the genome's median coverage). **c** Metapangenome construction. The environmental representation from **b** is then summarized for all genes and then overlaid onto the pangenome created in **a**—the inner layers show the genomic representation of the pangenome, and the outer layer shows the environmental representation of the pangenome. This outer layer summarizes the fraction of genes in each gene cluster that were environmentally core or accessory in those metagenomes (callout)

of the gene in the sample. Critically, this mapping of all samples to all genomes is naïve to any assumptions about which genomes occur in which habitats. The detection of a genome in a metagenome is operationally defined as the finding that at least half of the nucleotides in the genome are covered at least once. The coverage for each gene in a genome, for each of a large number of samples, can then be shown as a circular bar chart (Fig. 1b part 2), with concentric rings showing the coverage recruited from each metagenomic sample.

Comparative abundance in natural habitats can then be assessed for each gene using a metric to determine whether genes are core or accessory to an environment, rather than to a set of genomes [13]. This is accomplished by relating the abundance of a gene to the abundance of the genome that carries it, with respect to a set of environmental samples (Fig. 1b part 3). By this metric, a gene in an isolate genome is considered environmentally "core" if its median coverage, across all mapped metagenomes, is a given fraction of the median coverage of the genome in which it resides. We used a fraction of one-fourth, following the threshold used for metapangenomic analysis by Delmont and Eren [13]. The gene is environmentally "accessory" if its coverage falls below this cutoff. This method normalizes gene coverage to the genome and so is robust to differences in sequencing depth across samples. The one-fourth threshold is arbitrary, but most genes in our samples were either completely covered (detected) in many metagenomes and were environmentally core or recruited no coverage and were environmentally accessory (Additional file 1: Fig. S2, Supplemental Methods). Thus, the specific value of the core/accessory cutoff has minimal effect on the identification of genes as environmentally core or accessory. The environmental core/accessory metric provides a way of assessing the contribution of the gene to population structure in the environment—provided that the pangenome adequately represents the nucleotide sequences of genes found in the population. If the survival of a microbial cell in the environment under study depends on having this gene in its genome, the gene should register as environmentally core, while if the gene were dispensable or required in only a subset of the cells of the population, the gene would register as environmentally accessory.

The metapangenome (Fig. 1c) combines the pangenome with a summary of the mapping information. The outermost concentric ring of the metapangenome, here colored in red and blue, summarizes the environmental core/accessory metric across all genes in a gene cluster as a stacked bar chart with the heights normalized to the number of genes in that gene cluster. The scale of this outer ring thus changes from one part of the ring to another, as the number of genes per gene cluster ranges from one (as is the case between 10 o'clock and 12 noon on Fig. 1c) to three in the case of this example as shown between 3 o'clock and 8 o'clock on the figure. Thus, the metapangenome format summarizes the cultivar genome data in a visual format that emphasizes sets of shared or unique genes, and then summarizes the metagenomic data in the form of an environmental core/accessory metric for each of the genes in this pangenome, assessed across all the mapped metagenome samples.

### The apparent generalist H. parainfluenzae is composed of multiple subgroups

The species *H. parainfluenzae* is an apparent oral generalist in that it is both abundant and prevalent at multiple sites within the human mouth [7, 36]; however, previous

reports have suggested that genomically distinct sub-populations may exist within the mouth [7, 39]. To investigate the genome structure of the global *H. parainfluenzae* population as represented by the sequenced cultivated strains, we downloaded thirty-three high-quality isolate genomes from NCBI RefSeq. These genomes were sequenced over 8 years at 9 institutions with listed isolation sources ranging from sputum to blood (Additional file 2), with many from an unspecified body site. Thus, we consider it likely that each study and institution sampled from independent donors. We constructed a pangenome from these 33 genomes (Fig. 2, Additional file 3). Inspection of this pangenome (4318 gene clusters in total) shows a large core genome encompassing 35% of the pangenome ($N = 1493$ gene clusters), shown as the continuous black bars between 9 o'clock and 12 o'clock in Fig. 2. The dendrogram in the center of the figure organizes the gene clusters according to their presence/absence across genomes and thereby visually separates the core genome from the accessory genome. The accessory genome consists of 943 gene clusters (22% of the pangenome) unique to a single isolate genome, shown on the figure between 3 and 5 o'clock, and 1882 gene clusters (44% of the pangenome) shared by some but not all isolate genomes, shown between 5 o'clock and 9 o'clock on the figure. Functionally, while the core and accessory genome contained representatives of most COG categories, compositional differences were apparent, mostly due to fewer genes of unknown function in the core genome and fewer conserved functions like translation in the accessory genome (Additional file 1: Figure S3AB, Supplementary Text).

When the genomes themselves are clustered according to the number and identity of gene clusters that they share, they segregate into three groups (groups 1–3) that are distinguished by shared blocks of gene clusters (Fig. 2). The dendrogram in the upper right of the figure (Fig. 2) shows the clustering topography, and the major branch points in this dendrogram separate the groups. Genome completeness was > 99% and redundancy was < 10% in all genomes (Fig. 2, middle two gray bar charts), suggesting that the observed grouping is not based on the quality of the genome assemblies. As the number of gene clusters ranges from 1773 to 2071 per genome (Fig. 2, top right gray bar chart), the core of 1493 gene clusters represents 72–84% of the gene clusters in each genome and the gene clusters found in only a single genome contribute up to an additional 5%. Thus, collectively, the blocks of gene clusters that characterize the subgroups of *H. parainfluenzae* constitute a relatively small fraction of the genome.

### Haemophilus parainfluenzae subgroups are habitat-specific

Mapping of metagenomic data onto the genomes shows that the groups defined by genome content have significantly different distributions among oral sites ($p < 0.001$, permutational multivariate analysis of variance using Bray-Curtis dissimilarities, calculated using ADONIS in R [40]). We applied the competitive recruitment approach to the billions of short reads sequenced by the Human Microbiome Project (HMP [1, 2];) for hundreds of healthy individuals for three different oral habitats (tongue dorsum, TD; buccal mucosa, BM; and supragingival dental plaque, SUPP). The mapping information is summarized in the heatmap shown in the upper right corner of Fig. 2. For each oral habitat, the coverage of each genome by the median metagenome is displayed in the colored bar charts below the heatmap.

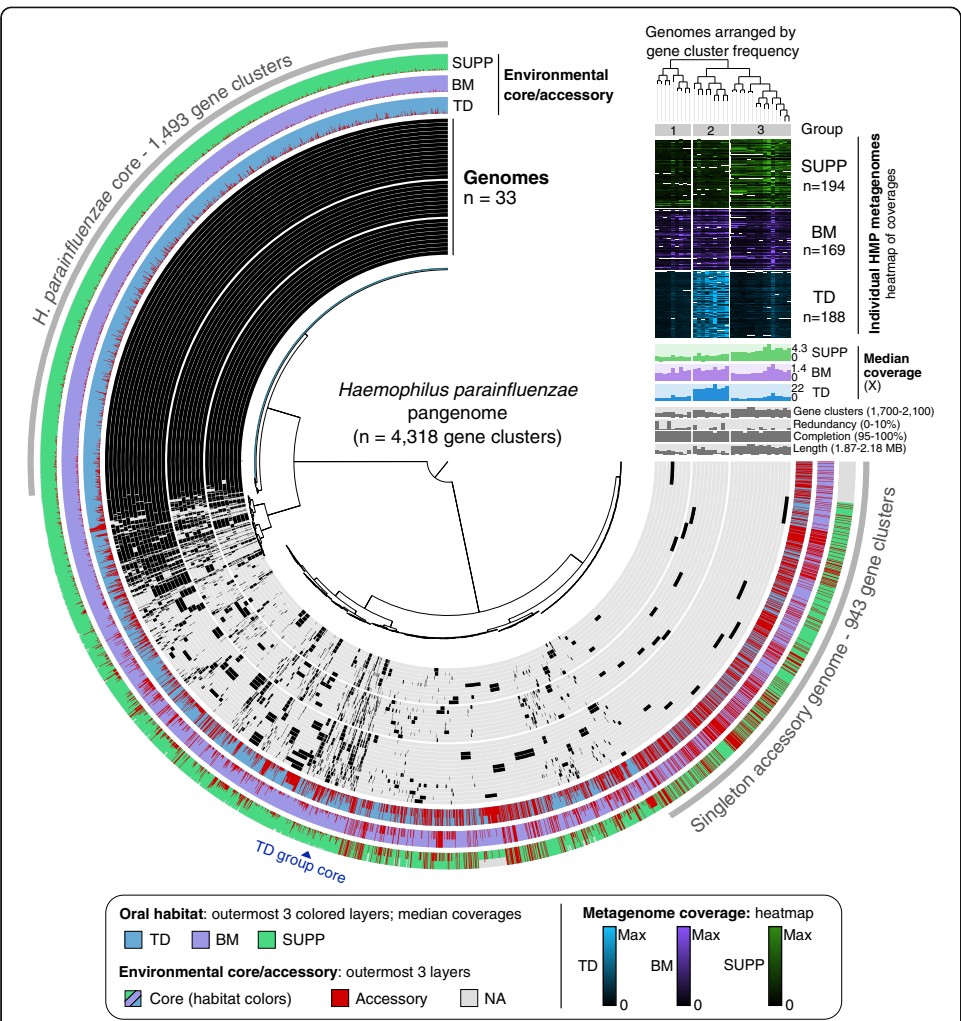

**Fig. 2** Metapangenomic analysis of *Haemophilus parainfluenza* reveals hidden diversity and habitat-specific subgroups. The inner radial dendrogram shows the 4318 gene clusters in the pangenome, clustered by presence/absence across genomes. The 33 genomes of *H. parainfluenzae* strains are plotted on the innermost 33 layers (black 270° arcs), spaced to reflect discernable groups based on genomic composition. Gene clusters within a given genome are filled in with black; gene clusters not present remain unfilled. Genomes are ordered by gene cluster frequency (top right dendrogram), with radial spacing added between major groups to improve visibility. The outermost three layers show the proportion of genes within each gene cluster determined to be environmental accessory (red) or core in HMP metagenomes from TD (blue), BM (violet), and SUPP (green), from inside to outside, respectively. If a genome was not well detected (< 0.5 of nucleotides covered by all metagenomes), all its genes were NA (gray) instead of environmentally core or accessory. Extending off the pangenome above 3 o'clock are bar charts of relevant information for each genome, with the y-axis limits in parentheses. Above the genome content summaries, each genome's median coverage across all TD, BM, and SUPP metagenomes is shown in the colored bar graph. Per-sample coverage of each genome is shown in the heatmap above, where each row represents a different sample, and cell color intensity reflects the coverage. Coverage is normalized to the maximum value of that sample (black = 0, bright = maximum; colors as before for each site)

Comparison of the pangenome groups with HMP coverage data shows that the middle group of genomes, group 2, is much more abundant in the 188 tongue dorsum metagenomes than genomes in the other two groups (Fig. 2 heatmap, median coverages). The heatmap in Fig. 2 shows that each TD metagenome typically provided high coverage to several group 2 genomes, although there was sample-to-sample variation in which genomes were most highly abundant. The median coverage bar plots show that

reads from the median TD metagenome covered each of the nine genomes in group 2 to an average depth of at least 15X, indicating that organisms similar to these strains are in high abundance on most people's tongues. Median coverage of the other twenty-four genomes by TD metagenomes is several-fold less (Fig. 2). By contrast, dental plaque metagenomes map with higher coverage to the genomes in group 3 (outermost group), whereas buccal mucosa metagenomes map with similar coverage to all three groups (Fig. 2). Thus, genomically defined subgroups of *H. parainfluenzae* have differential abundance across oral habitats, as reflected in differing levels of coverage of these genomes by metagenomic reads. Of these, the TD-abundant group of nine genomes appears most distinct.

Analysis of the phylogenetic relationships among *H. parainfluenzae* genomes, based on single-copy genes, revealed that groups defined by genome content differed from those defined by evolutionary relatedness at the strain level. We constructed a phylogeny based on nucleotide sequences from 139 bacterial genes previously identified as present in a single copy in most genomes [41]. This phylogeny placed the TD-associated genomes in separate clades of the *H. parainfluenzae* tree (Additional file 1: Fig. S4A). Two additional methods of assessing similarity, using 16S rRNA gene sequences and whole-genome kmer comparisons, provided little phylogenetic signal but indicated substantial nucleotide-level divergence among strains, respectively (Additional file 1: Fig. S4B). Thus, these analyses suggest that genomes of *H. parainfluenzae* that are enriched in tongue metagenomes share similar gene content but do not form a monophyletic evolutionary group.

### Genomic characteristics of the tongue-enriched H. parainfluenzae subgroup

Correspondence between genome content and environmental distribution raises the possibility that the success of a particular strain in a given habitat within the mouth may rely on the presence of certain genes fixed by selection. Specifically, we asked whether any genes were particularly characteristic of the nine *H. parainfluenzae* strains with high abundance in TD (Fig. 2, middle group of genomes). Only a small set of genes were present in all genomes of the TD group of *H. parainfluenzae* and not in any of the other isolate genomes; these genes are marked by a dark blue wedge labeled "TD group core" on the figure. We carried out a functional enrichment analysis, as described in Shaiber et al. [42], to compare the prevalence of predicted functions among TD genomes to their prevalence among non-TD genomes revealed by the metapangenome. This analysis identified exactly three functions in three gene clusters altogether encoding the three subunits of a sodium-dependent oxaloacetate decarboxylase enzyme exclusive to the TD group (Additional file 4). This enzyme converts oxaloacetate to pyruvate while translocating two sodium ions from the cytoplasm to the periplasm, providing a shunt to gluconeogenesis while establishing a potentially useful Na+ gradient [43, 44]. No complete oxaloacetate decarboxylase operon was detected in any of the other 24 *H. parainfluenzae* genomes (Additional file 1: Supplementary Text). No other functions or gene clusters had this universal presence in the TD-associated group but complete absence from the other *H. parainfluenzae* genomes. Aside from selection, an alternate explanation for the unique occurrence of this oxaloacetate operon in the TD-associated genomes could be shared evolutionary history, such as if these genomes were

all isolated from the same subject. However, not only are the TD-associated genomes not monophyletic (Additional file 1: Fig. S4A), they come from strains isolated from human sputum, the human toe, and the oropharynx of a rat and have sequences deposited by four different groups over 8 years (Additional file 2). Thus, the oxaloacetate operon stands out as a strong candidate for further experimental investigation into the source of selective advantage for the group of TD-abundant genomes in the tongue dorsum habitat.

Many *H. parainfluenzae* core gene clusters contain high proportions of gene sequences scored as environmentally accessory, particularly in TD (Fig. 2, shown as spikes of red in the "Environmental core/accessory" layer; Additional file 3). This result likely stems from differences in nucleotide-level sequence divergence from gene to gene within the population. These core gene clusters do contain sequences that are environmentally core to TD, i.e., the proportion of environmentally accessory sequences in these gene clusters is never 100% (Fig. 2). Thus, the traits represented by these core gene clusters are not missing from *H. parainfluenzae* living in the mouth. Further, metagenomic mapping can clearly distinguish between the genomes overall at the nucleotide level, as shown by the differential coverage results by habitat (Fig. 2 heatmap and median coverage bar chart). The differential abundance among some core genes' sequence variants thus suggests population-level differentiation between different oral habitats. As the pangenome contains proportionally fewer TD-representative genomes, the environmentally accessory gene sequences (red spikes) are higher in TD than in BM or SUPP. Although the metapangenome can identify gene sequences that are depleted in TD, it cannot discriminate between neutral and adaptive reasons for their differential abundance. Regardless, sequences for many *H. parainfluenzae* core genes are differentially present in certain habitats and may contain signatures of distinct subpopulations.

### Pangenomic analysis of oral members of the genus Rothia

Having decomposed the species *H. parainfluenzae* into discrete habitat-resolved subpopulations, we applied the same method of analysis to a genus, *Rothia*, that is composed of multiple habitat-specialized species [31, 36]. An advantage of constructing pangenomes at the genus level is that the genus-level core genes, as well as core and accessory genes of the individual species and strain groups, become readily identifiable. To assess the similarity of genome content among oral species as well as strains within the genus *Rothia*, we downloaded sixty-seven high-quality *Rothia* genomes from NCBI. Of the genomes for which the isolation source of the strain was reported, most were from sputum or bronchoalveolar lavage (Additional file 2). From these 67 genomes, we constructed a genus-level pangenome consisting of 5992 gene clusters (Fig. 3, Additional file 5).

One immediately evident feature of the oral *Rothia* genus pangenome is that the individual genomes segregate based on gene content into three major groups, each of which shares over 200–400 gene clusters that are absent from the others (Fig. 3). Taxonomic designations provided by NCBI (depicted by coloring the genome layers) show that these groups correspond to the three different recognized human oral *Rothia* species. A large set of gene clusters ($n = 1129$, 19% of the pangenome) were present in all

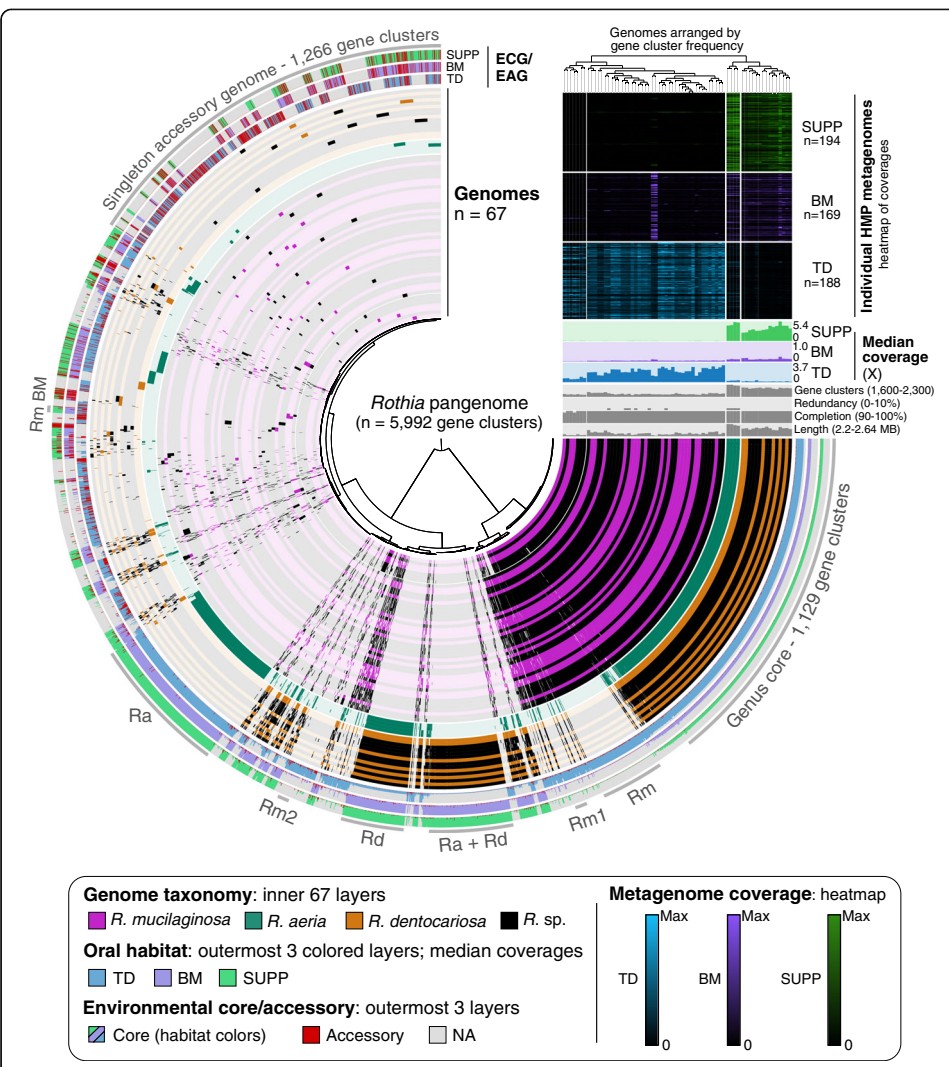

**Fig. 3** Genomes in the *Rothia* genus metapangenome are organized by gene content into groups that reveal associations to specific habitats. Tips on the inner radial dendrogram, setting the angular axis, correspond to gene clusters organized by presence/absence across genomes. The angular distance thus holds the entirety of the *Rothia* pangenome, 5992 distinct gene clusters. The inner 67 layers (270° arcs) represent genomes, colored by NCBI's taxonomic assignments, and are organized by their gene cluster frequencies (top right vertical dendrogram). Each genome's gene cluster content is displayed by filling in cells (gene clusters) for genomes in which that gene cluster is present. Gaps in radial spacing of layers delineate major groups determined by inspection of the pangenome and dendrogram. Groups of gene clusters are annotated with text and a gray arc—"Genus core" are gene clusters core to the genus *Rothia*; "Rm," *R. mucilaginosa*; "Rm1," *R. mucilaginosa* subgroup 1; "Ra + Rd," both *R. aeria* and *R. dentocariosa*; "Rd," *R. dentocariosa*; "Rm2," *R. mucilaginosa* subgroup 2; "Ra," *R. aeria*; "Rm BM," BM-abundant *R. mucilaginosa*. The outermost colored three layers show the proportion of genes within each gene cluster deemed environmental accessory (red) or core for HMP metagenomes from tongue dorsum (blue), buccal mucosa (violet), and supragingival plaque (green). If a genome was not well detected (< 0.5 of nucleotides covered by all metagenomes), all its genes were NA (gray) instead of environmentally core or accessory. Above the genome content summaries, each genome's median coverage across all TD, BM, and SUPP metagenomes is shown in the colored bar graph. Per-sample coverage of each genome is shown in the heatmap above, where each row represents a different sample, and cell color intensity reflects the coverage. Coverage is normalized to the maximum value of that sample (black = 0, bright = maximum; colors as before for each site)

of the *Rothia* genomes and represent the genus-level core genome. Given that each *Rothia* genome contains between 1693 and 2252 gene clusters, the genus core represents half to two-thirds of any given *Rothia* genome. Other sets of gene clusters were characteristic of and exclusive to *R. mucilaginosa* ($n = 207$, 3% of the pangenome), *R. dentocariosa* ($n = 274$, 5%), or *R. aeria* ($n = 455$, 8%). Taking the species-level core genes into account, the three *Rothia* species were identified unambiguously by their conserved gene content, with 77%, 77%, and 81% of the median *R. mucilaginosa*, *R. dentocariosa*, and *R. aeria* genomes, respectively, occupied by genus- or species-level core genes. The remaining ~ 20% of each genome represents accessory genes that were present in one or more genomes but not in all genomes of the species. Thus, for the genus *Rothia*, pangenomic analysis recapitulated species designations.

Genomes within a single group also form subgroups. The major group of *R. mucilaginosa* (Rm) genomes can be subdivided into two subgroups defined by 39 gene clusters exclusive to the larger subgroup (gray line "Rm1" in Fig. 3) and 38 gene clusters exclusive to the smaller subgroup (gray line "Rm2" in Fig. 3). Genomes deposited in NCBI with only a genus-level designation (i.e., *Rothia* sp.; black layers in Fig. 3) also fell into each *R. mucilaginosa* subgroup, increasing confidence in the discreteness of the subgroups. To investigate whether these gene clusters were localized to a single region, as could result from, e.g., a phage insertion, or whether they were scattered through the genome, we reordered the gene clusters (columns) to follow the genome order in an arbitrary *R. mucilaginosa* group 1 genome (Additional file 1: Fig. S5). The gene clusters present only in group 1 do not localize to a single region but are scattered throughout the chromosome (Additional file 1: Fig. S5), suggesting that the differentiation between the groups is not the result of a single recent chance event and may be instead the result of ecological and evolutionary pressures. Thus, *R. mucilaginosa* is comprised of at least two cryptic subgroups.

### Metagenomic mapping reveals habitat distributions of genome groups

Metagenomic mapping to the *Rothia* genomes demonstrated that the different genomic groups occupy different environments within the mouth. We carried out competitive mapping to the set of *Rothia* spp. cultivars using the same HMP metagenomic datasets as above. The resulting abundance information is summarized in the coverage heatmap and bar charts in Fig. 3. As in Fig. 2, the heatmap shows coverage data for hundreds of metagenomes (rows) collected from over a hundred different volunteers by the HMP. Two of the *Rothia* species (*R. aeria* and *R. dentocariosa*) were most abundant in SUPP samples, where the mean depth of coverage from the median SUPP metagenome was approximately 5X for the *R. aeria* genomes and 2 to 3X for most *R. dentocariosa* genomes (Fig. 3). The third species, *R. mucilaginosa*, was highly abundant in TD and for the most part displayed only negligible coverage from SUPP and BM samples. Outliers were also apparent—two genomes in the *R. mucilaginosa* group received high coverage from approximately one-third of BM metagenomes.

Whereas the heat map summarizes coverage information for each genome and metagenome as a single data point, the mapping analysis provides finer-grained information about the frequency of each gene sequence in natural populations by relating the abundance of each gene to the abundance of the genome that carries it. The three outer

multicolored layers of Fig. 3 summarize the outcome of this analysis with the *Rothia* genus pangenome for SUPP, BM, and TD samples. If a cultivar genome does not receive enough coverage in a habitat to be considered "detected," i.e., if more than half of a genome's nucleotides received no coverage in every metagenome from a habitat, then the result for genes from that genome is shown in gray rather than in color to indicate that the environmental core/accessory status could not be assessed.

Mapping results, as summarized by the environmental core/accessory metric, reinforced the conclusion from the coverage heatmap that the genomic groups corresponding to different named *Rothia* species occupied different habitats within the mouth. The *Rothia* genus core genes were environmentally core in all habitats except where their surrounding genome was not detected—which occurs because many of the *R. mucilaginosa* genomes were undetectable in BM and SUPP, and many of the *R. dentocariosa* genomes were undetectable in TD. In contrast, species-specific core genes were only environmentally core to specific habitats. The core genes unique to *R. mucilaginosa* ("Rm" gene clusters, Fig. 3) were environmentally core in TD, but their parent genomes were not detected in SUPP and BM—with the exception of the two *R. mucilaginosa* genomes that were abundant in BM and one that passed the detection threshold in SUPP (thin purple and green lines in Fig. 3). Conversely, the core genes unique to *R. dentocariosa* ("Rd," Fig. 3) were environmentally core in SUPP and BM, but their parent genomes for the most part were not detected in TD. Thus, these two species show distinct and complementary habitat distributions. *R. aeria* genomes behaved differently: they were detected in SUPP, BM, and TD and while their unique core genes ("Ra" gene clusters, Fig. 3) were environmentally core at all three sites, they attained significantly more coverage from SUPP than from TD or BM (Fig. 3, "median coverages") reflecting a bias towards SUPP. Thus, the core genes of *Rothia* species can distinguish their distinct habitat ranges. Investigating the predicted functions core to each species also supported the observed differentiation of species (Additional file 1: Supplementary Text, Fig. S3C).

### R. mucilaginosa genomes divide into subgroups

Subgroups can be distinguished within major species by the presence and absence of sets of gene clusters, and mapping of metagenomic reads can be used to assess whether these within-species groups have similar distribution patterns in the sampled oral habitats. The large group of *R. mucilaginosa* genomes can be subdivided at the pangenome level into two subgroups that differ by a small set of core genes ("Rm1" and "Rm2" in Fig. 3). Their genome-scale abundance as assessed by mapping is similar, with both recruiting coverage primarily from TD metagenomes (Fig. 3 heat map) and detected primarily in TD (Fig. 3 environmental core/accessory layers). Similarly, at the finer, gene level of mapping resolution, both the *R. mucilaginosa* group 1 core genes and the *R. mucilaginosa* group 2 core genes were environmentally core in TD. Further, both subgroups obtained high coverage from many HMP samples. Thus, these two *R. mucilaginosa* subgroups do not appear to be the result of a broad habitat shift such as from tongue to teeth or buccal mucosal sites. Instead, they may represent co-existing subpopulations, perhaps with distinct microhabitats within the TD community or between individual mouths.

We also detected evidence for habitat shifts between major habitats by a small number of genomes, in the form of outlier results in the mapping of human oral metagenomes to *Rothia* cultivar genomes. These outliers consist of the two *R. mucilaginosa* genomes that recruited high coverage from BM metagenomes (heat map, Fig. 3). The two *R. mucilaginosa* genomes abundant in BM satisfied the detection metric in BM and plaque, and the genes shared only by these two genomes (labeled RmBM in Fig. 3) were environmentally core in BM. This distribution provides evidence that a bacterium similar to *R. mucilaginosa* is in buccal mucosa samples at high enough abundance to satisfy the detection metric and that sequences from this bacterium find their closest match in these two *R. mucilaginosa* genomes in our set of sequenced cultivars.

### Two Rothia mucilaginosa genomes represent a BM-adapted subpopulation

To assess how well the outlier *R. mucilaginosa* genomes with high coverage and detection in BM represent a true buccal mucosa *Rothia* community, we inspected the coverage of one of the two outlier genomes, *R.* sp. E04, in more detail at the gene level (Fig. 4). In Fig. 4a, each unit around the near-complete circle represents a different gene in the genome, and the 90 small tracks show each gene's coverage in the 30 metagenomes per habitat with the highest *R.* sp. E04 coverage (Supplemental Methods). The BM and SUPP metagenomes covered the majority of this genome's genes relatively evenly, evidenced by the taller and more dense bars in the purple (BM) and green (SUPP) metagenomes, as expected for samples containing populations related to E04 (Fig. 4a). However, this pattern was not observed with TD metagenomes (Fig. 4a, inner 30 rings); instead, the coverage from TD metagenomes was low to absent in most regions of the genome and only dense for a handful of genes. Similar analysis of the other BM-abundant genome, *R.* sp. C03, revealed similar results (Additional file 1: Fig. S6). This pattern suggests the TD coverage results from spurious mapping of isolated regions of high similarity or mobility, e.g., phage elements. Particularly, the intermittent TD coverage at both the gene level (Fig. 4a) and the nucleotide level (Fig. 4b) indicates these samples do not contain a population closely related to *R.* sp. E04. If the reads from TD metagenomes that do map originate from phylogenetically related but non-*Rothia* or non-*R. mucilaginosa* populations, their evolutionary distance should produce higher densities of single-nucleotide variants relative to the *R.* sp. E04 genome. This is observed in Fig. 4b where the vertical black lines report the mean Shannon entropy of mapped nucleotide variants; high levels of entropy indicate variability at that nucleotide position. Thus, inspection of gene-level coverage of outlier isolate genomes validates these genomes as representatives of an *R. mucilaginosa* population more abundant in buccal mucosa rather than the tongue.

The identification of *R.* sp. E04 as the closest match to the BM-dwelling *R. mucilaginosa* population allows investigation into the genomic characteristics of the BM populations. There are 22 gene clusters shared by both genomes abundant in BM but absent from other known *Rothia* isolates (Fig. 3 genes "RmBM"). Further, these 22 gene clusters uniquely shared by *R.* sp. C03 and E04 are scattered throughout the genome (Fig. 4a genes marked with black tick marks, Additional file 1: Fig. S5), suggesting that these two genomes are not related simply by a single large shared insertion event, but that their set of distinctive genes accumulated over time. While these 22 gene clusters do

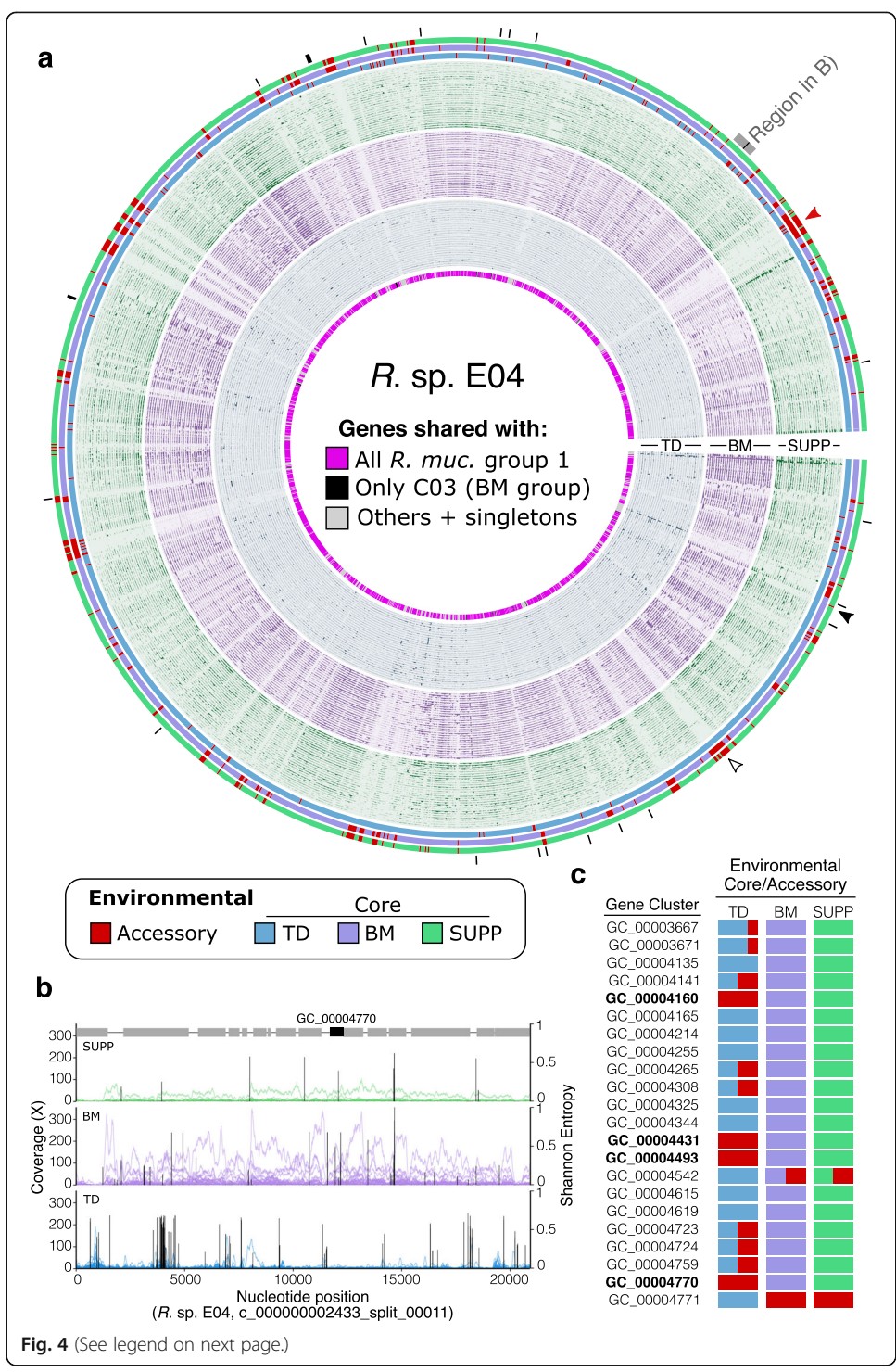

**Fig. 4** (See legend on next page.)

(See figure on previous page.)

**Fig. 4** Gene-scale metapangenomic analysis suggests candidate gene-level drivers of habitat adaptation. **a** Gene-level coverage of *Rothia* sp. E04. Units along the angular axis are *R*. sp. E04 genes, arranged in order found in *R*. sp. E04 with contigs joined arbitrarily. The innermost ring labels whether each gene was shared with all of *R. mucilaginosa* group 1 (pink), only between the BM-enriched strains *R*. spp. E04 and C03 (black; also shown with black lines outside figure), or otherwise (gray). The innermost 30 layers show coverage of each gene for 30 TD metagenomes with the highest coverage; middle 30, BM metagenomes; outer 30, SUPP metagenomes. Each layer's *y*-axis shows coverage by an individual sample, with *y*-axes scaled independently for each layer. The three outermost layers show whether genes were determined as environmental accessory (red) or core in TD (blue), BM (violet), or SUPP (green). Arrowheads show examples of gene abundance patterns: uniformly low-to-absent coverage across metagenomes (empty black) vs stochastically abundant but typically environmentally accessory (filled black). **b** Nucleotide-level coverage for a 20-kb contiguous stretch of *R*. sp. E04's genome that includes a candidate gene driver of the BM adaptation, GC_00004770. This stretch is shown by the labeled grey arc in **a**. Each trace shows a single sample's coverage, colored according to its oral site. Black bars show the mean Shannon entropy for variant sites covered at least 10x. Gray boxes above the SUPP traces mark genes, with GC_00004770 highlighted in black. **c** Table of the 22 gene clusters unique to *R*. sp. E04 and *R*. sp. C03 (also marked with black ticks in panel **a**). The columns labeled "Environmental Core/Accessory" show the fraction of genes in each gene cluster that are core (colored according to that habitat's color) or environmentally accessory (red). The corresponding Pfam function is listed for gene clusters for which a function could be predicted. The gene clusters environmentally core in BM and SUPP but not in TD are bolded

not contain any unique predicted functions, only a handful of them were environmentally accessory in TD but environmentally core elsewhere (bolded gene clusters in Fig. 4c). In other words, the four genes in bold in Fig. 4c recruited less coverage than their surrounding genomes in the 188 TD metagenomes where other *R. mucilaginosa* were abundant (Fig. 3 heatmap), yet these four genes had abundance similar to their surrounding genomes in the 169 BM and 198 SUPP metagenomes (Fig. 4a,b). These four genes thus represent prominent candidates for future experimental investigation of their potential role in the adaptation of a TD-abundant taxon to the new habitat of BM. However, even in BM and SUPP samples, large contiguous portions of the E04 and C03 genomes recruited little to no coverage relative to the rest of the genome (e.g., filled red arrowhead in Fig. 4a). Thus, although the cultivated strains *R*. sp. E04 and C03 are the best match out of all 69 genomes and provide some insight into the BM-inhabiting *R. mucilaginosa* population, the populations native to BM and possibly SUPP likely harbor many additional novel genes. In summary, gene-level mapping reveals features of the distribution of *Rothia* strains that suggest fine-tuned adaptation to each oral site.

**Some gene sequences are scarce across all metagenomes; others are intermittently abundant** More broadly, visualizing the mapping at the gene scale reveals different patterns of abundance for genes found in a single cultivated genome. Gene-level mapping highlights that some of this genome's sequences for both core and accessory genes may be at low abundance in the sample relative to the *R*. sp. E04 genome across the numerous metagenomes sampled. Many of the environmentally accessory genes are also accessory in the pangenome (genes from accessory gene clusters colored gray in the innermost ring, Fig. 4a), but many genes identified as members of the core genome (bright pink; innermost ring) are also environmentally accessory in HMP metagenomes. However, many such genes scored as 'environmentally accessory' are not uniformly low abundance across all individuals. Rather, they are at or below the limit of detection in

most samples (i.e., most mouths) but abundant in a few (e.g., the gene indicated by a filled black arrowhead in Fig. 4a). Thus, the existence of a cultivar containing this sequence indicates that cultivation selected a strain with this particular sequence from relatively low prevalence, either stochastically or as a result of cultivation bias. On the other hand, some gene sequences are more or less uniformly rare across samples (e.g., the empty black arrowhead in Fig. 4a). Such uniform rarity could reflect their restriction to a low-abundance niche or selection against that sequence in the environment, yet a lineage with that gene survived the bottleneck of isolation. Altogether, these results illustrate how a given cultivar's gene sequences may not be equivalently representative of environmental populations.

## Discussion

Metapangenomics provides a means of analyzing complex natural populations from a springboard of well-characterized isolate genomes. Starting from the solid foundation of virtually complete genome sequences from cultivated isolates, we constructed pangenomes and then used metagenomic read mapping to analyze the distribution of each gene in the pangenome in wild populations. Our metapangenomic analysis demonstrated that genomes that nominally belong to the same species in fact comprise habitat-specific subgroups within *H. parainfluenzae* and within a species of the genus *Rothia*.

Our metapangenomic findings elucidate the population structure of *H. parainfluenzae* and the genus *Rothia*, revealing differential distribution of species across habitats within the mouth that are within millimeters of one another and are in continual communication via saliva. Such biogeographic distributions have been suggested by prior studies based on cultivation as well as 16S rRNA gene surveys [32, 36, 38, 39]; however, the metapangenomic mapping approach is more comprehensive than cultivation and relies not on a marker gene to define a population but on complete genomes to demonstrate unequivocally the presence of different sets of microbes in the different habitats of supragingival plaque, tongue dorsum, and buccal mucosa.

The finding of habitat-specific subgroups shows that the buccal mucosa, in particular, is colonized by a previously unrecognized, distinctive microbiota. Among the three sampled oral habitats, dental plaque and the tongue dorsum are both characterized by extraordinarily dense, complex, and highly structured microbial communities [34, 38, 45, 46], whereas the buccal mucosa is more thinly colonized by a generally simpler set of microbes in which the genus *Streptococcus* comprises approximately half the cells. In this context, the *Rothia* spp. detected in 16S rRNA gene surveys on the buccal mucosa could be interpreted as microbes shed from the teeth and tongue and lodged transiently on the buccal mucosa or present in the sample as incidental contaminants from saliva. Our finding of a distinctive mapping pattern consistent across hundreds of HMP metagenomes for each of the three sampled oral sites rules out this interpretation and indicates that these microbes in fact represent a distinctive subpopulation adapted to the buccal mucosa niche.

Pangenomes that combine metagenome-assembled genomes (MAGs) with reference genomes can offer deeper insight into the gene pool and population structure of environmental microbes [47]. Given the rapid increase of publicly available MAGs [4, 48, 49], pangenomes of critical populations such as the BM-abundant *R. mucilaginosa* could be dramatically expanded. However, combining MAGs with cultivar genomes poses some

fundamental obstacles. Due to the inherent complexity of metagenome-sampled populations and assembly algorithms, a MAG is at best a consensus genome of closely related, perhaps clonal, members of a population, as opposed to a cultivar genome from a single cell's clonal lineage that provides comparatively higher confidence that all genes co-occur in the same cell [5, 8]. While assembly and binning algorithms are improving rapidly, poorly refined MAGs can suffer from significant contamination issues [6], which can influence ecological and pangenomic insights [50]. Hence, we focused only on high-quality cultivar genomes to benefit from higher confidence in their accuracy.

### Niche adaptation at the genomic level

Metapangenomic analysis reveals the differential abundance of genes across habitats and thus presents an opportunity to ask whether the presence or absence of particular genes is key to abundance in a given environment and whether these genes may encode traits under differential selection pressures. Several recent studies have applied functional enrichment or depletion analyses to a pangenome to investigate adaptation to the particular habitats from which those genomes were obtained [14, 15, 51]. Genomic biogeographic patterns also exist at the global scale, as shown by a recent study that identified differentiation in the motility and metabolic potential of European *E. rectale* populations relative to those from other continents [52]. Another recent study used methods conceptually similar to ours, i.e., using metagenomic abundances of reference genes to detect ecologically different subgroups within a population, to identify genes important for determining which *Bacteroides* strains engrafted from human mothers to infants [28]. By summarizing metagenomic recruitment across the entire pangenome, we extend such investigations of genomic biogeography to evolutionary scales, which allowed us to detect genomic subgroups with novel niche associations and directly investigate the frequencies of subgroup-specific genes among environmental populations. A limitation of the approach we used, however, is that it addresses only the presence and absence of coding sequences in the genome and cannot identify regulatory regions, structural variants, or other more subtle genomic traces of potential significance for niche adaptation.

Our finding of distinctive subpopulations of *H. parainfluenzae* is consistent with previous studies that have reported that *H. parainfluenzae* may be divided into at least three biotypes based on classical metabolic phenotyping [53]. Further, two recent studies employed population genetic approaches using metagenomes to detect *H. parainfluenzae* subpopulations with different habitat abundances [2, 39]. These and our results point towards this apparent generalist population being structured into distinct subpopulations that represent the partitioning of the oral niche. Additionally, the metapangenomic analysis we used identified the genomic subtypes and specific genes that are associated with the TD-abundant subpopulation, the oxaloacetate decarboxylase operon.

### The environmental relevance of reference genomes

Ideally, reference genomes for ecological analysis should accurately represent their native environments. However, in practice, most reference genomes are obtained from organisms that have been isolated and subjected to repeated subculture under laboratory

conditions. Many organisms are unable to grow under such conditions; those that grow may undergo genomic changes due to the imposition or relaxation of selection pressure under cultivation foreign to their native selective regime. Thus, it is important to evaluate the degree to which existing cultivar genomes serve as suitable references.

In general, although most core and some accessory genes of cultivars were well-represented in the oral environment, a few core and many accessory genes were uncommon in the oral environment. Genomic core and accessory genes do not correspond precisely to environmentally core and accessory genes, respectively. Particularly, singleton accessory genes were environmentally accessory, relative to their surrounding genomes. This result is not unexpected, as the set of accessory genes may be very large in an open pangenome whereas the microbiome in any mouth consists of a finite number of strains. Indeed, recent studies have shed light on the magnitude of previously unknown genes contained within the human microbiome [3, 4]. Nonetheless, it indicates that the features of any individual strain that are conferred by such accessory genes will be unrepresentative of a community in the mouth.

A major benefit of employing a metapangenomic strategy is that it permits us to identify which genes and cultivars are most representative of the microbiota growing in a given natural habitat. We used metapangenomics to query each gene from the *Rothia* and *H. parainfluenzae* cultivar pangenomes across metagenomes from the human oral cavity and measure the abundance of each cultivar gene across environments. These data can serve as a resource to guide the selection of the most environmentally representative strains and gene sequences for future experiments.

Cultivars are a valuable starting point for a metapangenomic analysis because they provide a high-quality foundation for assessing the presence, absence, and precise nucleotide sequence of genes. The source from which a cultivar is isolated, however, is not necessarily indicative of its environmental distribution; this distribution is more suitably assessed using metagenomic data. The Baas-Becking hypothesis that "everything is everywhere, but the environment selects" [54] suggests that the isolation of a single cell does not necessarily imply the existence of a population. The mapping of metagenomic data to a cultivar genome, by contrast, does indicate the overall abundance of an isolate in a habitat [42, 55], and the depth of coverage provided by different samples can indicate that the location of the highest abundance of a resident population may not be its original site of isolation. For example, the obligate bacterial symbiont TM7x was first isolated from a salivary sample in association with an *Actinomyces odontolyticus* strain [56]. However, as saliva is a transient mixture of bacteria shed from other oral sites, the ultimate source of TM7x remained ambiguous until metagenomic mapping was used to identify dental plaque as its native habitat [42]. Many of the genomes we used in this study came from strains isolated from sputum and non-oral sources such as blood, gallbladder, and skin (Additional file 2). Nonetheless, these genomes proved to be valuable references to probe the oral distribution of populations related to these genomes using metagenomic mapping. Based on our mapping results that show the high prevalence and abundance of oral populations similar to the isolate genomes, we infer that the strains isolated from blood and other non-oral samples are migrants dispersed from resident oral populations.

Mapping metagenomic short reads onto reference genomes can be used to investigate the relative divergence between a sampled population and the reference genome [17,

29]. The specific patterns of single-nucleotide variants (SNVs) among even closely related strains provide one of the most powerful ways to distinguish and track highly related strains, e.g., from mothers to infants [28]. In this study, we compared the relative frequencies of SNVs between different habitats as a proxy for relatedness to infer which sites had populations that were most similar to the reference strain. However, we did not explicitly search for specific SNVs that were enriched in one habitat vs. another. Future studies of nucleotide and codon variants across habitats will reveal the importance of nucleotide- and amino-acid-level changes for habitat specialization [57].

### We take no position on the species concept

Darwin, recognizing the difficulty in discriminating "doubtful species" [58], declined to discuss the various definitions that had been given to the term species in his day and indeed hoped that science would "be freed from the vain search for the undiscovered and undiscoverable essence of the term species." He noted that the amount of difference needed to confer the rank of species was at times "quite indefinite," that "no clear line of demarcation" could be drawn between species and sub-species, and therefore the term species was one "arbitrarily given for the sake of convenience" adding, " … to discuss whether they are rightly called species or varieties, before any definition of these terms has been generally accepted, is vainly to beat the air."

Metapangenomics does not alter Darwin's general conclusion; rather, it confirms and refines it at the genomic level. Our results indicate that for some taxa, such as the genus *Rothia*, pangenome analysis broadly supports the currently recognized species definitions, while for other taxa such as *H. parainfluenzae* habitat-associated subpopulations are detected that may or may not warrant species-level recognition. Even grouping whole genome sequences by hierarchic clustering based on gene composition results in "no clear line of demarcation." Rather, we observed a spectrum of genome cluster relatedness.

### Conclusions

In conclusion, our results reveal the detailed association between the environmental distribution and genomic diversity of oral bacterial populations. These patterns reveal that seeming generalist species are composed of cryptic subpopulations and that potentially only a small number of genes are associated with each subpopulation. More broadly, diversification to fully exploit available ecological niches is observed at many levels, from recognized species distinguished by many genes down to closely related subpopulations.

### Methods

Metapangenomes were prepared using publicly available genomes annotated as belonging to the genus *Rothia*, a gram-positive oral facultative anaerobe in the phylum Actinobacteria, and genomes annotated as the species *Haemophilus parainfluenzae*, a facultative gram-negative anaerobe in the phylum Proteobacteria. A flowchart linking the major methods and analyses is provided in Additional file 1: Fig. S1, and the Supplemental Methods, a detailed narrative methods document with reproducible code, is available at https://dutter.github.io/projects/oral_metapan.

### Genomic and metagenomic data acquisition

Genomes were downloaded from NCBI RefSeq based on the associated names using the assembly summary report obtained from ftp://ftp.ncbi.nlm.nih.gov/genomes/AS-SEMBLY_REPORTS/. Sequence names were simplified to contain only the strain identity and a unique contig id for all contigs, and then concatenated into a single FASTA file containing all genomes of interest. This file was used as the starting point for the anvi'o pangenomic analysis [20].

For the genus *Rothia*, 67 of the 73 genomes present in RefSeq were used. Genomes were inspected for potential errors and contamination which were identified based on expected genome size and gene count, fragmentary assemblies composed of short contigs, aberrantly high coverage of specific genes of unknown function (e.g., > 1000x coverage for genes that are neither rRNA nor mobile elements), and existing literature [59]. Of the six genomes not used, *R. nasimurium* was discarded for not being recognized as an oral resident by the HOMD, while *R.* sp. Olga and *R.* sp. ND6WE1A were discarded as non-oral isolates with aberrantly large unique gene contents (potentially contaminant genes). One *R. dentocariosa* genome was discarded for aberrant coverage and two *R. dentocariosa* genomes (OG2-1 and OG2-2) for containing potential contaminant genes based on aberrant coverage and for being identified as contaminated by Breitwieser et al. [59]. For *Haemophilus parainfluenzae*, all 33 genomes in RefSeq passed the contamination inspection and were used for analysis. Metadata from NCBI available for each strain is provided in Additional file 2.

Raw short-read metagenomic data from the Human Microbiome Project (HMP) were downloaded for tongue dorsum (TD, $n$ = 188), buccal mucosa (BM, $n$ = 169), and supragingival plaque (SUPP, $n$ = 194) using the HMP data portal at https://portal.hmpdacc.org/. These short-read data had undergone the HMP quality-control pipeline which includes trimming of low-quality bases and subsequently discarding of reads below 60 bp [1].

### Metapangenomic workflow

The metapangenome was constructed for each taxon using anvi'o with methods modified from Delmont et al. (2018; Additional file 1: Fig. S1). Open reading frames (ORFs) were identified on the downloaded contigs using Prodigal [60] and converted into an anvi'o-compatible database using the command anvi-gen-contigs-db. ORFs were exported and functionally annotated with InterProScan (version 5.30-69) using Pfam, TIGRFAM, ProDom, and SUPERFAMILY [61–64]. Coverage of each taxon's genomes was determined in each HMP metagenome using bowtie2 [65] with default parameters (--sensitive). Coverage of units encompassing multiple nucleotides, e.g., a gene or genome, is calculated from the per-nucleotide bowtie2 coverages by dividing the sum of nucleotide coverages in that unit by the number of nucleotides, as is the standard method for reporting a unit's coverage. Short reads from each metagenome were mapped against all genomes for that taxon; thus, the bowtie2 matched each read to the best-matching genomic locus, randomly choosing between multiple loci if they were equally best. Thus, coverage at highly conserved regions is affected by the total population abundance in that sample, while the coverage at variable loci reflects that particular sequence variant's abundance. These per-sample coverages were then incorporated

into the anvi'o database, and per-ORF coverages and summary metrics (max, min, mean, median) were determined. Single-nucleotide variants (SNVs) were also called per nucleotide if that nucleotide was covered at least 10x.

To compute pangenomes, we used the anvi'o workflow for pangenomics (see http://mer-enlab.org/p for a tutorial). Briefly, this workflow uses BLASTP [66] to compute amino-acid-level identity between all possible ORF pairs, from which removes weak matches by employing the --minbit criterion with default value 0.5 which requires that the bitscore of BLAST be at least half the maximum possible bitscore given the length of the sequences. The workflow then uses the Markov Cluster Algorithm (MCL) [67] to group ORFs into putatively homologous gene groups (gene clusters; Additional file 1: Supplementary Text) and aligns amino acid sequences in each gene cluster using MUSCLE [68] for interactive visualization. For display of the pangenome, the order of the genome layers was determined by clustering the genomes based on the frequency with which each gene cluster appeared in each genome, also shown as a dendrogram above the genome layers (e.g., top right of Figs. 2 and 3). Dendrogram branch length is fixed to an arbitrary constant.

Each gene's environmental core/accessory status was determined for any genomes covered at least 1x over half the genome length in at least one sample. A gene was classified as environmentally core with respect to an oral site if the median coverage of that gene by samples from that oral site was at least one-fourth the median coverage by those same samples of the genome from which it came [13]; otherwise, the gene was classified as environmentally accessory. If the genome was not confidently detected (if less than half of the nucleotides in the genome were covered) in any metagenome, all genes were reported NA instead of environmentally core or accessory.

A phylogenomic tree for the *H. parainfluenzae* isolates was constructed in RAxML [69] with the GTRCAT model and autoMRE boostrapping option using a nucleotide alignment of 139 concatenated single-copy core genes obtained with the anvi-get-sequences-for-hmm-hits program.

### Single-genome mapping

Per-sample coverage of a single genome was determined from the results of the meta-pangenomic analysis. For each oral habitat, coverage data for each strain's genome was extracted from the metapangenomic data using the command anvi-script-gen-distribution-of-genes-in-a-bin, with the same environmental detection threshold as above (0.25). Per-habitat analyses were combined using a custom wrapper script employing anvi'o functions, subsetting to show coverage for the 30 samples per oral habitat with the highest median coverages, while retaining the environmental core/accessory determination from all samples, not just the selected 30 samples (Supplemental Methods).

For visualization of nucleotide-level coverage in Fig. 4b, coverage was obtained using the anvi-get-split-coverages for the splits containing the gene(s) of interest and plotted with a custom R script (Supplemental Methods). SNV information from each metagenome was reported using anvi-gen-variability-profile command (variability information was recorded during the mapping step described earlier) which outputs the Shannon entropies of each variable position. Higher Shannon entropy signifies more environmental variability while low entropy signifies less variability. The mean of the observed Shannon entropies from all reporting metagenomes was then plotted for each position by oral habitat.

## Functional enrichment analyses

Functional enrichment analyses were carried out following the pipeline described in Shaiber et al. [42]. The analysis uses the anvi-get-enriched-functions-per-pan-group function, which de-replicates the predicted functions of each genome and then carries out a series of functional contrasts between specified groups of genomes. For the *H. parainfluenzae* analysis, the three groups displayed in Fig. 2 were used as the groups, and TIGRFAM functions were used. The analysis identifies enriched and depleted functions by group based on the prevalence of each function in genomes belonging to that group vs. the prevalence of that function in genomes outside that group.

Figures were generated with anvi'o [20] and cleaned for publication in Inkscape (https://inkscape.org/).

## Supplementary Information

---

**Additional file 1.** Supplementary text and Figures.

**Additional file 2.** Accession information and metadata for genomes used in this study. Genomes are listed in the same order as in Figs. 2 and 3, from inside to outside.

**Additional file 3.** Summary of *H. parainfluenzae* gene clusters. Each row in the table describes a different gene, listing the genome from which it came, the gene cluster to which it belongs, its predicted function, and other summary information.

**Additional file 4.** Functional enrichment results for *H. parainfluenzae*. Each row reports the enrichment of a TIGR FAM function in a group or groups of *H. parainfluenzae* genomes, according to the groups labeled in Fig. 2.

**Additional file 5.** Summary of *Rothia* gene clusters. Each row in the table describes a different gene, listing the genome from which it came, the gene cluster to which it belongs, its predicted function, and other summary information.

**Additional file 6.** Functional enrichment results for *Rothia* species. Each row reports the enrichment of a different TIGRFAM function in one or more *Rothia* species.

**Additional file 7.** Review history.

**Additional file 8: Table S1.** Number of gene clusters per pangenome with varying MCL inflation factors.

---

### Acknowledgements

We thank Floyd Dewhirst for helpful discussions. We thank the many researchers who deposited the publicly available genomes we used. We also thank the anonymous reviewers for their comments that resulted in an improved publication. The content is solely those of the authors and does not necessarily reflect the views of Harvard Catalyst, Harvard University and its affiliated academic health care centers, the National Science Foundation, or the National Institutes of Health.

### Peer review information

### Review history

The review history is available as Additional file 7.

### Authors' contributions

D.R.U., G.G.B., A.M.E., C.M.C., and J.L.M.W. designed the research; D.R.U., G.G.B., and J.L.M.W. performed the research and analyzed the data; D.R.U., G.G.B., A.M.E., C.M.C., and J.L.M.W. wrote the paper. The authors read and approved the final manuscript.

### Authors' information

Twitter handles: @bajenak (Daniel R. Utter) and @JMarkWelch (Jessica L. Mark Welch).

### Funding

Support was provided to D.R.U, C.M.C, and G.G.B from Harvard Catalyst | The Harvard Clinical and Translational Science Center (National Center for Research Resources and the National Center for Advancing Translational Sciences, National Institutes of Health Award UL1 TR001102 and financial contributions from Harvard University and its affiliated academic health care centers). Additional support was provided from NIH Grant DE022586 to G.G.B. Additional support was provided by the National Science Foundation Graduate Research Fellowship Program under Grant No. DGE1745303 to D.R.U. Additional support was provided to D.R.U by Harvard University's Department of Organismic and Evolutionary Biology program.

### Availability of data and materials

The raw data used in this study are publicly available at NIH GenBank and RefSeq (https://www.ncbi.nlm.nih.gov/genome/) for genomes (specific genome accessions listed in Additional file 2) and HMP metagenomes from https://portal.hmpdacc.org/. Analyzed data in the form of anvi'o databases can be found at https://doi.org/10.6084/m9.figshare.11591763 [70]. A reproducible methods document hosted at https://dutter.github.io/projects/oral_metapan [71] provides the code used for the analyses.

### Ethics approval and consent to participate

Not applicable

### Consent for publication

Not applicable

### Competing interests

The authors declare that they have no competing interests.

### Author details

[1]Department of Organismic and Evolutionary Biology, Harvard University, Cambridge, MA 02138, USA. [2]The Forsyth Institute, Cambridge, MA 02142, USA. [3]The Josephine Bay Paul Center for Comparative Molecular Biology and Evolution, Marine Biological Laboratory, Woods Hole, MA 02543, USA. [4]Department of Medicine, University of Chicago, Chicago, IL 60637, USA.

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

## 
