## [**Additional file 7.** Review history. · Genome Biology]

Review History

First round of review

Reviewer 1

Are you able to assess all statistics in the manuscript, including the appropriateness of statistical tests used? Yes, and I have assessed the statistics in my report.

Comments to author:

Genome Biology Manuscript Number: GBIO-D-20-00130

Title: Metapangenomics of the oral microbiome provides insights into habitat adaptation and cultivar diversity.

In this reviewer's opinion what makes this work highly relevant and appealing to readers of genome biology is its attempt to partial out the pangenome from a core genome, leveraging both metagenomics and previously published whole genome sequences. The authors use the oral microbiome as a compelling system of study. To facilitate the use of their workflow by others, the authors include a reproducible script, which is highly commendable. However, failing inclusion of additional information this manuscript falls short of broad appeal to a general audience.

As it is currently written, the authors do not focus enough on the details that make this paper of high general interest. Specifically, in the supplemental text, where discussion of pangenome construction is confined, much of the information is ellipted. For example, paragraph 1 ends with: Thus, differentiating between a legitimately unique gene cluster vs an over-split gene cluster is of importance when doing comparative genomics. This statement leaves the reviewer with the question, how does one differentiate between legitimately unique clusters and over-split clusters? What data did the authors present to suggest 25% is legitimate? As a second example, in the second paragraph, "Despite varying inflation parameters, the resultant gene clusterings were qualitatively the same and quantitatively similar." Is it possible to have a supplemental figure to demonstrate this? There are many more examples in the supplemental text. As one reads through the entirety of the supplemental text one is left with similar questions. Further, nowhere can I find summaries of general descriptions of the extent to which we see genes partition into accessory/core clusters, and how this compares to other attempts to work on pangenomes. The more the authors can elaborate on the logic that was used to produce the paper, supporting that logic with data and fully discussing the data presented, the higher the impact the paper. One way to orchestrate this would be to integrate much of the presentation of figures, as currently presented in methods, into the supplemental text. Otherwise, this reads as an article that would be of more interest to readers in a domain specific journal focused on oral microbiology.

In addition, the authors make the argument that they are observing habitat "preference" and "adaptation". However, they do not provide evidence against competing hypotheses. Throughout, this reviewer is left to wonder whether the genomes that segregate across sites do so because they reflect historical contingencies, perhaps are artifacts of inter-individual variation, as one alternative. Further, nowhere in the text does this reviewer see specific tests demonstrating adaptation. Failing the inclusion of such tests, it's important to use more appropriate language.

Specific concerns are noted below:

1. Overall I think Figure 1 nicely outlines the thrust of the computational workflow.
2. Line 120-130: Where does the $\frac{1}{4}$ threshold come from for classification of core/accessory? Is it necessary to make this a binary variable for the purpose of defining core/accessory? Please justify this metric. As articulated, it seems entirely arbitrary. If you were to use this as a quantitate trait with quantitate coloring it could seem to function as a confidence score rather than a yes/no, on/off trait.
3. In the discussion of the metapangenome, lines 130-140, in the HMP dataset, how many strains are you dealing with for the key taxa in your paper from an individual mouth, with your assemblies, how does that compare to what was published, how many co-occurring genomes do you have within each

mouth? Throughout the text of this paper, I find myself wondering whether you're seeing an artifact of historical effects from founders in a handful of individuals rather than seeing habitat selection. I think figure 1 should take multiple strains from single mouths into account in some way or present this information as supplemental figures.

4. Figure 2: I cannot judge this figure in full since the figure isn't high enough resolution for me to zoom in on it and read the text. These comments are based on what I can see.

5. Lines 145-150: Subsequent to this paper from the Salzberg group (Human contamination in bacterial genomes has created thousands of spurious proteins, *Genome Research*, 2019. doi:10.1101/gr.245373.118), any group that downloads Refseq genomes for use as the heart of the paper, must demonstrate that the gene/protein content reflects non-contaminant sequences. Having contaminated genomes as the reference can introduce distortions in your estimates of core/accessory components. Please outline specific methods used for QC of refseq data and HMP data downloads.

6. Lines 150-160: During discussion of this region, I'm finding it hard to keep track of the totals. For example, is it possible to say: Inspection of this pangenome shows a large core genome encompassing 75% of X gene clusters (N= 1,493 gene clusters) and to make similar changes throughout the text to help the reader understand the various scales of comparison? In any case, please state a total before dividing into accessory and core clusters.

7. Here again, I think it's important to convince the reader that we're not seeing a cluster of core genes for the TD because of a handful of individuals, or strains for one individual over time, who were sampled in the HMP cohort. Please for this and other cases where an argument is being made for habitat selection provide evidence against historical contingency having occurred in a handful of mouths.

8. Figure 2: Can you kindly walk the reader through the interpretation of the gene clusters, redundancy, completion and length graphs in the figure legend? If they contribute meaningfully to interpretation it may be nice to include a sentence or two in the supplemental discussion or in the main text.

9. Figure 2: There seems to be a peak that is core across all habitats, Supp, BM, TD. Can you comment on what is included here as core in the supplement, if not the main text? What is shared is just as important as what is different when discussing variability between habitats.

10. Line 165: This reviewer is not familiar with a statistical test called Adonis which is suitable for use in this context. Can a citation for the method be provided?

11. Additional file 3: Can you please mark the habitat from which the other genomes derived on this figure? Given that an argument is being made about "habitat selection" it's important in virtually all figures for the reader to be able to discern habitats of origin. Also, can the text be clarified to indicate to indicate the extent to which these clades map to genomes from different individuals, and at different time points?

12. Line 185ish: It's a bit strange to read that "they don't share a unique common ancestor to the exclusion of other strains" - that language reads like a figurative speedbump. It's important to keep in mind you're looking at 9 genomes and we don't have any context about where the comparator genomes came from.

Metagenomic mapping reveals habitat preferences of genome groups

13. Figure 3: This figure also needs to have improved resolution; it's hard to see much of the finer details

14. Lines 192-196: It is a bit Lamarckian to claim that the presence of these genes in the genome is responsible for their success in a given habitat. Can this be reframed to reflect environmental selection of genes leading to their observed enrichment?

15. Lines 206-210: "No other functions or gene clusters evidenced this universal presence in the TD-associated group but the complete absence from the other *H. parainfluenzae* genomes". Again, please rule out the possibility that something in the metadata including for example interpersonal differences explains this.

16. Line 220: The differential abundance among some core genes' sequence variants thus suggests population-level differentiation between different oral habitats. Again, please rule out the possibility that something in the metadata explains this

17. Line 230, insert reference for claim that Rothia has multiple habitat specialists
18. Line 230ish: For Rothia same quality control for excluding contaminant regions should be articulated in methods
19. Line 236-"Segregate into 3 major groups each of which shares a substantial set of genes that are absent from the others": can you articulate the 3 major groups precisely?
20. Line 290, can it be said they show distinct habitat distributions rather than preferences?
21. No data have been presented to suggest that these taxa wouldn't be detected across habitats - do we know that we have sampled the metagenome of each habitat to completion (richness)? Do we see that these distributions arise when considering multiple (and most) individuals in the dataset rather than just a subset of donors? To me, these considerations need to be argued, in the form of supplemental figures, before making the conclusion that you're observing habitat selection.
22. Figure 4A: I'm having a hard time linking the text to my reading of the figure - is it possible to refine the text so that readers who aren't used to looking at plots like this one can easily digest the figure while reading the main text?
23. Lines 350-355: Is it right to say that you're demonstrating adaptation to a habitat? Other than distributional differences in presence/absence, which can be due to limited depth of sequencing, differential depths based on habitat and historical contingencies, what evidence has been provided in support of adaptation?
24. Conclusions: please modify these if tests for adaptation are not included in revision.
25. Line 500: Can you please summarize for the reader the habitats from which these Rothia genomes were sequenced? Awareness of the originating habitat is important for this work and shouldn't be relegated to a supplementary data file for the reader to snoop through independently. The same is true for the Haemophilus genomes.
26. Lines 535-580: Figures and data shouldn't be presented in the methods. Summarize the workflow in the methods and rework this text into the supplemental text, integrating it so it doesn't seem like a separate person wrote it, so the reader can understand this information as it is presented in the paper.
27. Line 575: When is this companion paper going to be published? If not before this one, then a summary of the methods used to do the functional enrichment should be included. Otherwise, the methods presented in this paper cannot stand alone.

Reviewer 2

Are you able to assess all statistics in the manuscript, including the appropriateness of statistical tests used? No, I do not feel adequately qualified to assess the statistics.

Comments to author:

The manuscript "Metapangenomics of the oral microbiome provides insights into habitat adaptation and cultivar diversity" employed a pangenome approach to study the population structure in three sites of oral habitat.

Major strength:

1. The meta-pangenome pipeline and transparent protocol presented in this paper is a big strength - a great demonstration of combining metagenome and genomes to characterize niche-specific genomic features.
2. One interesting aspect is how to combine metagenome and genomes. Previous studies have suggested to combine metagenome-assembled genomes and genomes, while this study uses genomes of isolates and reads-mapping of metagenomes onto these genome references - a novel implementation.
3. Both species-pan and genus-pan were included.

Major weakness:

1. The rationale for choosing *H. parainfluenzae* and *Rothia* is unclear. Just 2 exemplar taxa? How are they able to represent the population structure to study habitat niche adaptation of oral microbiome? Or they are the only "habitat generalists" in oral microbial community?
2. One of the biggest advantages of reads-mapping approach to combine metagenomes is to quantify the sequence variants' abundance so to study population structure, which was only briefly mentioned but not discussed in this study. Any insight or discussion on this point?
3. One of the major observations of this study is the limited capability of cultivated organisms to represent population in an environment, which is not new.
4. Any genomic characteristics of the *Rothia* habitat-specialized species/subspecies/subgroups, such as functional groups? While there are number of gene clusters as unique/core, no discussion on functional aspect was included?

We thank the reviewers for their close reading of the manuscript and for their positive comments and expert critique. Both reviewers remarked favorably on the experimental design of combining metagenomes with information from sequenced isolate genomes. Reviewer 1 found this element of the work to be “highly relevant and appealing to readers of genome biology” and “the oral microbiome as a compelling system of study.” Reviewer 2 found that “The meta-pangenome pipeline and transparent protocol presented in this paper is a big strength” and that characterizing niche-specific genomic features using read-mapping onto isolates was “a novel implementation.”

The reviewers also had numerous suggestions for improvement of the manuscript, which we address point by point below. Reviewer 1, in particular, asked for more discussion and data concerning the logic used to produce the paper, in order to make clear the relevance, interest, and impact of the paper for a general audience. We agree with this request and in response we have added three additional supplementary figures and one supplementary table. The first two figures provide data supporting the logic used to produce our paper, and one remaining figure and table report new analyses comparing gene functions between the core and accessory genome as well as inter-species comparisons for the genus *Rothia*. We also streamlined the methods section and reworked the supplemental text as requested by the reviewer.

Here we respond to individual points raised by the reviewers, with reviewer comments in blue text and our responses in black text.

Reviewer #1:

In this reviewer's opinion what makes this work highly relevant and appealing to readers of genome biology is its attempt to partial out the pangenome from a core genome, leveraging both metagenomics and previously published whole genome sequences. The authors use the oral microbiome as a compelling system of study. To facilitate the use of their workflow by others, the authors include a reproducible script, which is highly commendable. However, failing inclusion of additional information this manuscript falls short of broad appeal to a general audience.

As it is currently written, the authors do not focus enough on the details that make this paper of high general interest. Specifically, in the supplemental text, where discussion of pangenome construction is confined, much of the information is ellipped. For example, paragraph 1 ends with: Thus, differentiating between a legitimately unique gene cluster vs an over-split gene cluster is of importance when doing comparative genomics. This statement leaves the reviewer with the question, how does one differentiate between legitimately unique clusters and over-split clusters? What data did the authors present to suggest 25% is legitimate? As a second example, in the second paragraph, "Despite varying inflation parameters, the resultant gene clusterings were qualitatively the same and quantitatively similar." Is it possible to have a supplemental figure to demonstrate this? There are many more examples in the supplemental text. As one reads through the entirety of the supplemental text one is left with similar questions. Further, nowhere can I find summaries of general descriptions of the extent to which we see genes partition into accessory/core clusters, and how this compares to other attempts

to work on pangenomes. The more the authors can elaborate on the logic that was used to produce the paper, supporting that logic with data and fully discussing the data presented, the higher the impact the paper. One way to orchestrate this would be to integrate much of the presentation of figures, as currently presented in methods, into the supplemental text. Otherwise, this reads as an article that would be of more interest to readers in a domain specific journal focused on oral microbiology.

We thank the reviewer for these excellent suggestions. Our specific revisions for each suggestion are provided under the relevant point-by-point “Specific Concerns” below.

To address the reviewer’s broader suggestion to clarify the supporting logic presented in the Supplemental Text and demonstrate the impact of parameter choices on the data, we have added a supplemental figure, Supplemental file 11, and substantially revised the opening paragraphs (Lines 1253-1278) of the Supplemental Text to read as follows:

“A crucial element underlying construction of a pangenome is being able to identify and group homologous genes. Within a group of closely related genomes, amino acid sequences of homologous genes are likely to be largely conserved across genomes while non-homologous genes both within and across genomes are distinct. Thus, clusters for a species- or genus-level pangenome may be unambiguous. Nonetheless, ambiguous homology and errors in clustering may occur. We used two methods to investigate the overall robustness and validity of our homology definitions – first, determining the robustness of the pangenome to various amino acid similarity thresholds, and second, assessing the level of functional heterogeneity within our gene clusters.

Our pangenome construction approach compares amino acid sequences for all gene pairs, prunes weak hits, and resolves the network of hits with the Markov Cluster Algorithm (MCL) to determine gene clusters. MCL uses a hyperparameter, “inflation,” to adjust the clustering sensitivity, i.e., the tendency to split clusters. To gauge robustness of the pangenome to the inflation parameter of the MCL algorithm, we varied the inflation parameters by ± 2 . The resultant number of gene clusters was quantitatively similar (Supplementary Table 1), differing by <0.5% for *H. parainfluenzae* and <2.5% for *Rothia*, and the pangenome arrangement was qualitatively similar in that the overall pattern and relative size of the genus core (in the case of *Rothia*), species cores, and accessory genome remained nearly identical (Additional file 11).

Gene clusters are defined purely by amino acid sequence similarity. Although functional similarity is not part of the definition, nevertheless, intuitively one expects to produce gene clusters that are composed of genes with similar function. We assessed the validity of this expectation by assessing the fraction of gene clusters whose constituent genes were annotated with different COG functions. Heterogeneity of functional annotation within a gene cluster was rare in our data; for *H. parainfluenzae*, only 2.6% (75 out of 2892 gene clusters with predicted COG functions) of gene clusters had within-cluster functional heterogeneity, and *Rothia* was comparably low at 3.5% (96 of 2757 gene clusters with COG annotation). “

In addition, the authors make the argument that they are observing habitat "preference" and "adaptation". However, they do not provide evidence against competing hypotheses. Throughout, this reviewer is left to wonder whether the genomes that segregate across sites do so because they reflect historical contingencies, perhaps are artifacts of inter-individual variation, as one alternative. Further, nowhere in the text does this reviewer see specific tests demonstrating adaptation. Failing the inclusion of such tests, it's important to use more appropriate language.

We thank the reviewer for making this important point. We have revised throughout to either use more appropriate language or provide specific reasoning for why adaptation is the observation vs. inter-individual variation, drift, or historical contingency. Our specific revisions are provided under the relevant point-by-point concerns below.

In general, the Background now introduces alternative hypotheses and their relevance to the oral microbiome, and these hypotheses are subsequently addressed with data in the results. The results and supporting data are more clearly stated to communicate how the patterns observed are consistent across hundreds of individuals and as such are unlikely to be a result of historical contingencies.

Specific concerns are noted below:

1. Overall I think Figure 1 nicely outlines the thrust of the computational workflow.

Thank you for this kind comment.

2. Line 120-130: Where does the $\frac{1}{4}$ threshold come from for classification of core/accessory? Is it necessary to make this a binary variable for the purpose of defining core/accessory? Please justify this metric. As articulated, it seems entirely arbitrary. If you were to use this as a quantitative trait with quantitative coloring it could seem to function as a confidence score rather than a yes/no, on/off trait.

We appreciate the opportunity to clarify this important point. We have added a supplemental figure showing that most genes are either completely detected in the majority of samples or completely undetected, therefore the exact threshold for ECG/EAG has little importance for most genes. We also revised the text to indicate that our choice of $\frac{1}{4}$ as a threshold follows the first paper using the method with the same threshold. The section (Lines 143-151) now reads:

“ ... a gene in an isolate genome is considered environmentally “core” if its median coverage, across all mapped metagenomes, is a given fraction of the median coverage of the genome in which it resides. We used a fraction of one-fourth, following Delmont & Eren (2018). The gene is environmentally “accessory” if its coverage falls below this cutoff. This method normalizes gene coverage to the genome and so is robust to differences in sequencing depth across samples. The one-fourth threshold is arbitrary, but most genes in our samples were either completely covered (detected) in many metagenomes and were environmentally core, or recruited no coverage and were environmentally accessory (Additional File 2, Supplemental Methods). Thus, the specific

value of the core/accessory cutoff has minimal effect on the identification of genes as environmentally core or accessory.”

We agree with the reviewer that Interpreting accessory/core as a continuous variable is informative. We use this more nuanced view of the environmental distribution of genes in our analysis of the single-genome views of coverage, e.g., Lines 494-512 where we report and interpret the distribution of individual gene coverages across samples. For the overall summary metric, however, if the data are maintained as a quantitative trait, they cannot readily be summarized onto the pangenome at the gene cluster level. At this overview level it would be necessary to combine potentially different values for each gene in a gene cluster. Thus, we chose to make the metric produce a binary accessory/core for the purposes of inclusion in the pangenome, but we fully agree and support the interpretation of the underlying data as a continuous variable as well, e.g. Figure 4A, which shows the abundance of each gene in each metagenome as a continuous variable.

3. In the discussion of the metapangenome, lines 130-140, in the HMP dataset, how many strains are you dealing with for the key taxa in your paper from an individual mouth, with your assemblies, how does that compare to what was published, how many co-occurring genomes do you have within each mouth? Throughout the text of this paper, I find myself wondering whether you're seeing an artifact of historical effects from founders in a handful of individuals rather than seeing habitat selection. I think figure 1 should take multiple strains from single mouths into account in some way or present this information as supplemental figures.

We thank the reviewer for raising the important point of historical contingency. We are slightly confused about the reviewer's points about HMP data, strains per mouth, etc. in relation to this section and Figure 1, as this is a simplified cartoon describing the method. Our revisions and response here assume the reviewer is commenting on these topics in association with Figure 2, please correct us if that is not the reviewer's intention.

For the reviewer's first question about how many co-occurring genomes we find per mouth, all genomes are searched for in the same pool of samples, thus each genome may or may not co-occur in any mouth's community sampled by the metagenome. We have clarified this in the manuscript, as well as our operational threshold for determining if a genome is detected or not in a metagenome (>0.5 horizontal coverage) by adding a sentence reading (Lines 133-136):

“Critically, this mapping of all samples to all genomes is naïve to any assumptions about which genomes occur in which habitats. The detection of a genome in a metagenome is operationally defined as the finding that at least half of the nucleotides in the genome are covered at least once.”

We also added a sentence (Lines 218-220) to the paragraph initially describing the *H. parainfluenzae* mapping to report that most TD-group genomes were abundant in each TD metagenome (mouth):

“The heatmap in Figure 2 shows that each TD metagenome typically provided high coverage to several Group 2 genomes, although there was sample-to-sample variation in which genomes were most highly abundant.”

For the second question about historical contingency, the observed differential distribution is a repeated pattern across hundreds of mouths, which does not support a hypothesis of historical contingency or founder effects. We have clarified this fact in the manuscript by editing the introduction (Lines 80-95) to read:

“The microbiomes that assemble in the different oral habitats are clearly related to one another – composed of many of the same genera, for example – but are largely composed of different species (Mark Welch et al. 2019). For example, the major oral genera *Actinomyces*, *Fusobacterium*, *Neisseria*, *Veillonella*, and *Rothia* occur throughout the mouth, but their individual species show strongly differential habitat distributions. Individual species within these genera typically have 95-100% prevalence across individuals and make up several percent of the community at one oral site, but show lower prevalence and two orders of magnitude lower abundance at other oral sites (Eren et al., 2014; Mark Welch et al., 2016; Mark Welch et al., 2019; Wilbert et al., 2020). The reproducibility of taxon distribution across individuals, despite the frequent communication of the habitats with one another via salivary flow, suggests that founder effects and other stochastic processes are unlikely to explain the differences in species-level distribution and that these differences likely arise from selection. However, some apparent “habitat generalist” species, such as *Haemophilus parainfluenzae*, *Streptococcus mitis*, and *Porphyromonas pasteri*, can be found throughout the mouth (Eren et al., 2014; Segata et al., 2012). Altogether, the mouth is colonized by well-characterized bacteria that build distinctive communities in the different oral habitats in the absence of dispersal barriers.”

Throughout the results we have also added language clarifying the logic that the repeatability of species abundance patterns across hundreds of independent samples suggests that historical contingency is not a likely cause for the observed patterns.

4. Figure 2: I cannot judge this figure in full since the figure isn't high enough resolution for me to zoom in on it and read the text. These comments are based on what I can see.

We regret that file compression during uploading may have impacted the figure quality. We will ensure that no loss occurs during the resubmission process. To ensure readability during the review process, we also made available high-resolution versions of our figures here that allows anonymous access:

<https://drive.google.com/drive/folders/1VkkX-QfVeJdHUPYMu6hzoXOjLJ3HgPdC?usp=sharing>

5. Lines 145-150: Subsequent to this paper from the Salzberg group (Human contamination in bacterial genomes has created thousands of spurious proteins, Genome Research, 2019. doi:10.1101/gr.245373.118), any group that downloads Refseq genomes for use as the heart of the paper, must demonstrate that the gene/protein content reflects non-contaminant sequences. Having

contaminated genomes as the reference can introduce distortions in your estimates of core/accessory components. Please outline specific methods used for QC of refseq data and HMP data downloads.

This is a critical consideration and we appreciate the reviewer's care in this matter. We have updated the methods to explicitly state the QC criteria used during genome selection and refined the genome list by combining our criteria with the Salzberg supplementary material. The Salzberg group identified only one *R. dentocariosa* genome as containing human contamination and no *Haemophilus parainfluenzae* genomes; similarly, our approach flagged and discarded an additional 4 *Rothia* genomes besides the Salzberg-identified *Rothia* genome, but no *H. parainfluenzae*. The methods section describing the genome selection now reads (Lines 694-704):

"Genomes were inspected for potential errors and contamination which were identified based on expected genome size and gene count, fragmentary assemblies composed of short contigs, aberrantly high coverage of specific genes of unknown function (e.g., >1000x coverage for genes that are neither rRNA nor mobile elements), and existing literature (Breitwieser et al. 2019). Of the six genomes not used, *R. nasimurium* was discarded for not being recognized as an oral resident by the HOMD, while *R. sp. Olga* and *R. sp. ND6WE1A* were discarded as non-oral isolates with aberrantly large unique gene contents (potentially contaminant genes). One *R. dentocariosa* genome was discarded for aberrant coverage and two *R. dentocariosa* genomes (OG2-1 and OG2-2) for containing potential contaminant genes based on aberrant coverage and for being identified as contaminated by Breitwieser et al. (2019). For *Haemophilus parainfluenzae*, all 33 genomes in RefSeq passed the contamination inspection and were used for analysis."

The HMP metagenomes available at HMPDACC have already undergone basic QC according to HMP parameters, and we have updated the methods (Lines 708-710) to summarize the HMP QC steps:

"These short-read data had undergone the HMP quality-control pipeline which includes trimming of low-quality bases and subsequently discarding of reads below 60bp (HMP Consortium, 2012)."

6. Lines 150-160: During discussion of this region, I'm finding it hard to keep track of the totals. For example, is it possible to say: Inspection of this pangenome shows a large core genome encompassing 75% of X gene clusters (N= 1,493 gene clusters) and to make similar changes throughout the text to help the reader understand the various scales of comparison? In any case, please state a total before dividing into accessory and core clusters.

We thank the reviewer for this excellent suggestion. The section (Lines 180-181) now reads:

"Inspection of this pangenome (4,318 gene clusters in total) shows a large core genome encompassing 35% of the pangenome (N= 1,493 gene clusters), ..."

We have gone through the manuscript to ensure similar introduction of gene cluster totals and percentages.

7. Here again, I think it's important to convince the reader that we're not seeing a cluster of core genes for the TD because of a handful of individuals, or strains for one individual over time, who were sampled in the HMP cohort. Please for this and other cases where an argument is being made for habitat selection provide evidence against historical contingency having occurred in a handful of mouths.

We thank the reviewer for this important point regarding historical contingency, which we have addressed in response to the reviewer's question 3 (see above). Although the genomes' metadata do not allow individual-specific tracking to confidently identify their origins, the genomes span 8 years and 9 different institutions based on available NCBI metadata. Thus, while we cannot prove that donors were not the same between studies and institutions, it is likely that many or most of the genomes are independent. We have updated the results section (Lines 175-179) to clarify the isolates' diverse backgrounds:

"... we downloaded thirty-three high-quality isolate genomes from NCBI RefSeq. These genomes were sequenced over 8 years at 9 institutions with listed isolation sources ranging from sputum to blood (Additional File 3), with many from an unspecified body site. Thus, we consider it likely that each study and institution sampled from independent donors."

Also, we have revised the section reporting that certain genomes were more abundant in TD to more clearly state that our claim is based on the abundances of these genomes in 188 different HMP metagenomes and therefore not based on a handful of mouths (Lines 216-218):

"Comparison of the pangenome groups with HMP coverage data shows that the middle group of genomes, Group 2, is much more abundant in the 188 tongue dorsum metagenomes than genomes in the other two groups (Figure 2 heatmap, median coverages)."

8. Figure 2: Can you kindly walk the reader through the interpretation of the gene clusters, redundancy, completion and length graphs in the figure legend? If they contribute meaningfully to interpretation it may be nice to include a sentence or two in the supplemental discussion or in the main text.

We appreciate this suggestion and have updated the main text as follows to refer to these graphs (Lines 198-203):

"Genome completeness was >99% and redundancy was <10% in all genomes (Figure 2, middle two grey bar charts), suggesting that the observed grouping is not based on the quality of the genome assemblies. As the number of gene clusters ranges from 1,773 to 2,071 per genome (Figure 2, top right grey bar chart), the core of 1,493 gene clusters represents 72-84% of the

gene clusters in each genome and the gene clusters found in only a single genome contribute up to an additional 5%.”

9. Figure 2: There seems to be a peak that is core across all habitats, Supp, BM, TD. Can you comment on what is included here as core in the supplement, if not the main text? What is shared is just as important as what is different when discussing variability between habitats.

We thank the reviewer for this suggestion. The second reviewer also suggested a similar analysis specifically for the *Rothia* results, and to address both reviewers' suggestions we have added a new section to the supplemental text reporting these results with a new supporting figure and references in the text. The *H. parainfluenzae* portions are as follows:

Introduced in main text (Lines 191-194):

“Functionally, while the core and accessory genome contained representatives of most COG categories, compositional differences were apparent, mostly due to fewer genes of unknown function in the core genome and fewer conserved functions like translation in the accessory genome (Additional File 5AB, Supplemental Text).“

Supplemental text paragraphs lines 1367-1401 (subheading “**Functions of the core and accessory genome for *H. parainfluenzae* and *Rothia***“):

“In addition to comparing differences between genomes based on gene content, we also investigated functional differences between core and accessory genes and between species of *Rothia* and strains of *H. parainfluenzae*.

To investigate functional similarities and differences between core and accessory genes, we assessed the frequencies of each COG category in core, singleton accessory, and intermediate accessory genes as identified based on the pangenome. For simplicity we compared only genes assigned a single COG category and omitted genes that were assigned multiple COG categories. For *H. parainfluenzae*, the core consisted of gene clusters shared by all 33 genomes; the singleton accessory genome, gene clusters found in exactly one genome; and the intermediate accessory genome, gene clusters occurring in 2-32 genomes. Overall, each portion of the pangenome contained genes belonging to each COG category (Additional File 5A) but the frequencies differed. For example, genes involved in translation (J) and nucleotide metabolism (F) were both more numerous and proportionally more enriched in the core genome. On the other hand, defense mechanisms (V) and the mobilome (X) were more abundant in both the singleton and the intermediate accessory genome.

To investigate functional enrichment in one set of genomes compared to another, we recorded the proportion of genomes containing each TIGRFAM function. From this proportional data, the enrichment of each function in each group was determined using a logistic regression by the method of Shaiber et al. (2020). The full enrichment data is presented in Additional File 7 for

each gene. To obtain a high-level view of which group(s) were more similar based on shared functions, we aggregated the enrichment scores by subtracting the mean proportional occurrence of each function in the group(s) in which it was not enriched from the mean of its proportional occurrence in the group(s) in which the TIGRFAM was enriched (Additional File 5B). For example, if a function was enriched in Groups 1 and 2 with a proportional occurrence of 1 and 0.8 in Groups 1 and 2 but also 0.1 in Group 3, the aggregate enrichment would be $(0.8 + 1)/2 - 0.1 = 0.8$. This aggregate enrichment of each function is shown in Additional File 5B. The three genes of the oxaloacetate operon unique to Group 2 stand out clearly, but more broadly the functional similarity between groups can be estimated. Group 2 and Group 3 share more genes with higher enrichment than do Group 1 and Group 2, or Group 1 and Group 3. This observation agrees with the arrangement of genomes based on gene cluster content shown as the dendrogram arranging genome layers in Figure 2, which places Group 2 sister to Group 3.”

10. Line 165: This reviewer is not familiar with a statistical test called Adonis which is suitable for use in this context. Can a citation for the method be provided?

We thank the reviewer for this suggestion and have changed the line to include the category of statistic (permutational MANOVA) as well as a reference to Anderson 2001. Line 208 now reads:

“(p < 0.001, permutational multivariate analysis of variance using Bray-Curtis dissimilarities, calculated using ADONIS in R; Anderson, 2001).”

11. Additional file 3: Can you please mark the habitat from which the other genomes derived on this figure? Given that an argument is being made about "habitat selection" it's important in virtually all figures for the reader to be able to discern habitats of origin. Also, can the text be clarified to indicate the extent to which these clades map to genomes from different individuals, and at different time points?

We appreciate this comment, and we have updated Additional File 9 to document the isolation source associated with each genome to help provide this. However, the reviewer’s comment touches upon a larger conceptual point we would like to clarify: that mapping metagenomes onto pangenomes reveals the natural residence of a microbial population more confidently than can the sample origin of a strain. We have added a new Discussion paragraph to this end. The paragraph (Lines 630-648)reads:

“Cultivars are a valuable starting point for a metapangenomic analysis because they provide a high-quality foundation for assessing the presence, absence, and precise nucleotide sequence of genes. The source from which a cultivar is isolated, however, is not necessarily indicative of its environmental distribution; this distribution is more suitably assessed using metagenomic data. The Baas-Becking hypothesis that “everything is everywhere, but the environment selects” (Baas-Becking, 1934) suggests that the isolation of a single cell does not necessarily imply the existence of a population. The mapping of metagenomic data to a cultivar genome, by contrast, does indicate the overall abundance of an isolate in a habitat (Kraal et al., 2014; Shaiber et al.,

2020), and the depth of coverage provided by different samples can indicate that the location of highest abundance of a resident population may not be its original site of isolation. For example, the obligate bacterial symbiont TM7x was first isolated from a salivary sample in association with an *Actinomyces odontolyticus* strain (He et al., 2015). However, as saliva is a transient mixture of bacteria shed from other oral sites, the ultimate source of TM7x remained ambiguous until metagenomic mapping was used to identify dental plaque as its native habitat (Shaiber et al. 2020). Many of the genomes we used in this study came from strains isolated from sputum and non-oral sources such as blood, gallbladder, and skin (Additional File 3). Nonetheless these genomes proved to be valuable references to probe the oral distribution of populations related to these genomes using metagenomic mapping. Based on our mapping results that show the high prevalence and abundance of oral populations similar to the isolate genomes, we infer that the strains isolated from blood and other non-oral samples are migrants dispersed from resident oral populations.”

12. Line 185ish: It's a bit strange to read that "they don't share a unique common ancestor to the exclusion of other strains" - that language reads like a figurative speedbump. It's important to keep in mind you're looking at 9 genomes and we don't have any context about where the comparator genomes came from.

We agree with the reviewer that these lines are confusing. We have revised the line as follows to clarify that the preceding analyses focused on the genomes found to be enriched in tongue metagenomes, but evolutionary relatedness cannot fully explain their similarities (Lines 253-255):

“Thus, these analyses suggest that genomes of *H. parainfluenzae* that are enriched in tongue metagenomes share similar gene content but do not form a monophyletic evolutionary group.”

Metagenomic mapping reveals habitat preferences of genome groups

13. Figure 3: This figure also needs to have improved resolution; it's hard to see much of the finer details

We regret that file compression during uploading may have impacted the figure quality. We will ensure that no loss occurs during the resubmission process. To ensure readability during the review process, we also made available high-resolution versions of our figures here that allows anonymous access: <https://drive.google.com/drive/folders/1VkkX-QfVeJdHUPYMu6hzoXOjLJ3HgPdC?usp=sharing>

14. Lines 192-196: It is a bit Lamarckian to claim that the presence of these genes in the genome is responsible for their success in a given habitat. Can this be reframed to reflect environmental selection of genes leading to their observed enrichment?

We appreciate the suggestion and have so reworded the sentence (Lines 257-259) to now read:

“Correspondence between genome content and environmental distribution raises the possibility that the success of a particular strain in a given habitat within the mouth may rely on the presence of certain genes fixed by selection.”

15. Lines 206-210: "No other functions or gene clusters evidenced this universal presence in the TD-associated group but the complete absence from the other *H. parainfluenzae* genomes". Again, please rule out the possibility that something in the metadata including for example interpersonal differences explains this.

We thank the reviewer for this clarification. We have revised the section as follows to report how the available metadata does not suggest any features that could explain why these 9 genomes would be grouped together (Lines 284-291):

“No other functions or gene clusters had this universal presence in the TD-associated group but complete absence from the other *H. parainfluenzae* genomes. Aside from selection, an alternate explanation for the unique occurrence of this oxaloacetate operon in the TD-associated genomes could be shared evolutionary history, such as if these genomes were all isolated from the same subject. However, not only are the TD-associated genomes not monophyletic (Additional File 6A), they come from strains isolated from human sputum, the human toe, and the oropharynx of a rat and have sequences deposited by four different groups over 8 years (Additional File 3).”

16. Line 220: The differential abundance among some core genes' sequence variants thus suggests population-level differentiation between different oral habitats. Again, please rule out the possibility that something in the metadata explains this

We appreciate this important point. Please see our revisions in response to concern # 15 above where we address the metadata for these same genomes.

17. Line 230, insert reference for claim that *Rothia* has multiple habitat specialists

We appreciate this suggestion and have added two references supporting this claim, Segata et al. 2012 and Mark Welch et al. 2019. Line 319 now reads:

“... we applied the same method of analysis to a genus, *Rothia*, that is composed of multiple habitat-specialized species (Mark Welch et al., 2019; Segata et al., 2012).”

18. Line 230ish: For *Rothia* same quality control for excluding contaminant regions should be articulated in methods

We thank the reviewer for their attention to controls. Please see our added methods details in response to question 5 above.

19. Line 236-"Segregate into 3 major groups each of which shares a substantial set of genes that are absent from the others": can you articulate the 3 major groups precisely?

We now describe the characteristics defining the groups in the first sentence (Lines 326-330):

"One immediately evident feature of the oral *Rothia* genus pangenome is that the individual genomes segregate based on gene content into three major groups, each of which shares over 200-400 gene clusters that are absent from the others (Figure 3). Taxonomic designations provided by NCBI (depicted by coloring the genome layers) show that these groups correspond to the three different recognized human oral *Rothia* species."

20. Line 290, can it be said they show distinct habitat distributions rather than preferences?

We thank the reviewer for this important distinction and fully agree. Lines 405-406 now read

"... distinct and complementary habitat distributions"

We have also changed the section header to similarly read 'distributions' instead of preferences – **"Metagenomic mapping reveals habitat distributions of genome groups"**.

21. No data have been presented to suggest that these taxa wouldn't be detected across habitats - do we know that we have sampled the metagenome of each habitat to completion (richness)? Do we see that these distributions arise when considering multiple (and most) individuals in the dataset rather than just a subset of donors? To me, these considerations need to be argued, in the form of supplemental figures, before making the conclusion that you're observing habitat selection.

We assume this comment is referring to the first paragraph of our section "Metagenomic mapping reveals habitat distribution of genome groups" and are revising as such. Please correct us if this is not the case.

For the reviewer's first point, assessing sampling to completion is a fundamentally challenging task for metagenomes as their true composition is variable and unknown, and the existing metagenomes seem to only sequence the surface of human microbial diversity (see Figure 6 of Tierney et al. 2019 Cell Host & Microbe <https://doi.org/10.1016/j.chom.2019.07.008>). However, we have taken the following steps to ensure that differential sequencing depth has minimal impact on our results:

First, the selection of *Rothia* and *H. parainfluenzae* as exemplar taxa was not random – please see our response to comment 1 of Reviewer 2 for our rationale for their inclusion and for our edits to the manuscript to address this point. Briefly, both taxa are known from previous studies to be both prevalent and abundant, making them more likely to be present in the metagenomes than rarer taxa may be.

Also, the data shown is normalized when possible to ameliorate variable sequencing depth. In the heatmaps of Figures 2 and 3, the coverage data shown in each row (sample) is normalized to that sample's maximum coverage of any genome. So, for example, in the heatmap where a given SUPP row (metagenome) is dark for the *R. mucilaginosa* genomes but bright green for some *R. dentocariosa* genomes, then in that SUPP metagenome the *R. dentocariosa* genomes had high coverage while *R. mucilaginosa* genomes had little to no coverage. To clarify this normalization, we have edited the figure legends (e.g., Lines 1090-1091) to read:

“... each row represents a different sample, and cell color intensity reflects the coverage. Coverage is normalized to the maximum value of that sample”

For the author's second point about whether the observed patterns hold across multiple individuals, we are revising the text to explicitly clarify that the heatmaps shows the nearly 200 metagenomes sampled per habitat. As the HMP collected at most 3 metagenome samples per donor, the observed results do come from many different individuals are not a result of subsetting to a small number of donors. The text now reads (Lines 367-370):

“The resulting abundance information is summarized in the coverage heatmap and bar charts in Figure 3. As in Figure 2, the heatmap shows coverage data for hundreds of metagenomes (rows) collected from over a hundred different volunteers by the HMP.”

We suspect that the compression of the figure during the upload process obscured the annotations in Figures 2 and 3 stating the large N for each habitat. We are uploading a higher resolution during the resubmission and are enlarging the text to make sure the number of samples is readily apparent from these figures.

22. Figure 4A: I'm having a hard time linking the text to my reading of the figure - is it possible to refine the text so that readers who aren't used to looking at plots like this one can easily digest the figure while reading the main text?

We appreciate the opportunity to refine this section and we have added several clarifying details. The text introducing this figure now reads (Lines 436-446):

“To assess how well the outlier *R. mucilaginosa* genomes with high coverage and detection in BM represent a true buccal mucosa *Rothia* community, we inspected the coverage of one of the two outlier genomes, *R. sp. E04*, in more detail at the gene level (Fig. 4). In Figure 4A, each unit around the near-complete circle represents a different gene in the genome, and the 90 small tracks show each gene's coverage in the 30 metagenomes per habitat with the highest *R. sp. E04* coverage (Supplemental Reproducible Workflow). The BM and SUPP metagenomes covered the majority of this genome's genes relatively evenly, evidenced by the taller and more dense bars in the purple (BM) and green (SUPP) metagenomes, as expected for samples containing populations related to E04 (Fig. 4A). However, this pattern was not observed with TD

metagenomes (Figure 4A, inner 30 rings); instead the coverage from TD metagenomes was low to absent in most regions of the genome and only dense for a handful of genes.”

23. Lines 350-355: Is it right to say that you're demonstrating adaptation to a habitat? Other than distributional differences in presence/absence, which can be due to limited depth of sequencing, differential depths based on habitat and historical contingencies, what evidence has been provided in support of adaptation?

We thank the reviewer for the opportunity to clarify the basis of our claim. The statement is based on the consistent pattern from the hundreds of HMP metagenomes from different mouths, and so the pattern is robust to historical contingency (please see responses above to the reviewer's questions 3, 7, 11, 15, and 21 about historical contingency). Based on the reviewer's excellent comment earlier, we have clarified the text when we introduce the environmental accessory / core designation that the method is normalized to the genome's coverage, thus accounting for differential sequencing depths between samples and habitats. We have revised the text (Lines 481-484) to more clearly state this basis.

“In other words, the four genes in bold in Figure 4C recruited less coverage than their surrounding genomes in the 188 TD metagenomes where other *R. mucilaginoso* were abundant (Figure 3 heatmap), yet these four genes had abundance similar to their surrounding genomes in the 169 BM and 198 SUPP metagenomes (Figure 4A,B).”

24. Conclusions: please modify these if tests for adaptation are not included in revision.

We have modified the Conclusion (Lines 675-680) to avoid the word “adaptation” as follows:

“In conclusion, our results reveal the detailed association between the environmental distribution and genomic diversity of oral bacterial populations. These patterns reveal that seeming generalist species are composed of cryptic subpopulations and that potentially only a small number of genes are associated with each subpopulation. More broadly, diversification to fully exploit available ecological niches is observed at many levels, from recognized species distinguished by many genes down to closely related subpopulations.”

25. Line 500: Can you please summarize for the reader the habitats from which these *Rothia* genomes were sequenced? Awareness of the originating habitat is important for this work and shouldn't be relegated to a supplementary data file for the reader to snoop through independently. The same is true for the *Haemophilus* genomes.

We thank the reviewer for raising this important point. Please see our response to the reviewer's question 11 where we have added a paragraph to the discussion describing the benefits of inferring originating habitat from metagenomes rather than isolation source, and our response to the reviewer's question 7 in which we now describe in the text the isolation source for the *H. parainfluenzae* genomes.

We have also added to the text a sentence describing the isolation source of the *Rothia* genomes. This section now reads (Lines 322-325):

“... we downloaded sixty-seven high-quality *Rothia* genomes from NCBI. Of the genomes for which the isolation source of the strain was reported, most were from sputum or bronchoalveolar lavage (Additional File 3). From these 67 genomes we constructed a genus-level pangenome ...”

26. Lines 535-580: Figures and data shouldn't be presented in the methods. Summarize the workflow in the methods and rework this text into the supplemental text, integrating it so it doesn't seem like a separate person wrote it, so the reader can understand this information as it is presented in the paper.

We have moved the details and results to the supplementary text, and revised the methods (Lines 750-752) to read as follows:

“The workflow then uses the Markov Cluster Algorithm (MCL) (van Dongen & Abreu-Goodger, 2012) to group ORFs into putatively homologous gene groups (gene clusters; Supplemental Text), ...”

27. Line 575: When is this companion paper going to be published? If not before this one, then a summary of the methods used to do the functional enrichment should be included. Otherwise, the methods presented in this paper cannot stand alone.

The companion paper is in revision and also has been posted on bioRxiv. We have updated the references (Lines 1008-1010) to refer to the bioRxiv preprint:

“Shaiber, A., Willis, A. D., Delmont, T. O., Roux, S., Chen, L.-X., Schmid, A. C., et al. (2020). Functional and genetic markers of niche partitioning among enigmatic members of the human oral microbiome. *bioRxiv*, 2020.04.29.069278.”

Specific Comments from Reviewer #2:

The manuscript "Metapangenomics of the oral microbiome provides insights into habitat adaptation and cultivar diversity" employed a pangenome approach to study the population structure in three sites of oral habitat.

Major strength:

1. The meta-pangenome pipeline and transparent protocol presented in this paper is a big strength - a great demonstration of combining metagenome and genomes to characterize niche-specific genomic features.
2. One interesting aspect is how to combine metagenome and genomes. Previous studies have suggested to combine metagenome-assembled genomes and genomes, while this study uses genomes

of isolates and reads-mapping of metagenomes onto these genome references - a novel implementation.

3. Both species-pan and genus-pan were included.

We appreciate the reviewer's kind words.

Major weakness:

1. The rationale for choosing *H. parainfluenzae* and *Rothia* is unclear. Just 2 exemplar taxa? How are they able to represent the population structure to study habitat niche adaptation of oral microbiome? Or they are the only "habitat generalists" in oral microbial community?

We thank the reviewer for mentioning this important point. We have revised the introduction to introduce *Rothia* and *H. parainfluenzae* in the context of the other oral taxa (Lines 80-95):

“The microbiomes that assemble in the different oral habitats are clearly related to one another – composed of many of the same genera, for example – but are largely composed of different species (Mark Welch et al. 2019). For example, the major oral genera *Actinomyces*, *Fusobacterium*, *Neisseria*, *Veillonella*, and *Rothia* occur throughout the mouth, but their individual species show strongly differential habitat distributions. Individual species within these genera typically have 95-100% prevalence across individuals and make up several percent of the community at one oral site, but show lower prevalence and two orders of magnitude lower abundance at other oral sites (Eren et al., 2014; Mark Welch et al., 2016; Mark Welch et al., 2019; Wilbert et al., 2020). ... However, some apparent “habitat generalist” species, such as *Haemophilus parainfluenzae*, *Streptococcus mitis*, and *Porphyromonas pasteri*, can be found throughout the mouth (Eren et al., 2014; Segata et al., 2012).”

We have also updated the introduction with the following lines (99-106) to clearly explain our logic behind choosing these taxa as complementary examples:

“We focused on two exemplar oral taxa with high prevalence (>95%; Segata et al. 2012, Eren et al. 2014, Mark Welch et al. 2016) and high abundance in the mouth: the species *Haemophilus parainfluenzae* and the genus *Rothia*. These two taxa represent the two oral biogeographic patterns, with *H. parainfluenzae* representing apparent habitat generalist species and the genus *Rothia* representing genera composed of habitat-specific species. Both *Rothia* spp. and *H. parainfluenzae* are sufficiently abundant – making up on average several percent of the microbiota at their sites of highest abundance – that metagenomic read recruitment to reference genomes can reliably sample their natural populations.”

2. One of the biggest advantages of reads-mapping approach to combine metagenomes is to quantify the sequence variants' abundance so to study population structure, which was only briefly mentioned but not discussed in this study. Any insight or discussion on this point?

We appreciate this excellent suggestion and have added the following discussion paragraph comparing our use of sequence variant frequency to estimate population similarity with other SNV-tracking approaches (Lines 659-657):

“Mapping metagenomic short reads onto reference genomes can be used to investigate the relative divergence between a sampled population and the reference genome (Simmons et al., 2008; Deneff 2018). The specific patterns of single-nucleotide variants (SNVs) among even closely-related strains provide one of the most powerful ways to distinguish and track highly-related strains, e.g., from mothers to infants (Yassour et al., 2018). In this study, we compared the relative frequencies of SNVs between different habitats as a proxy for relatedness to infer which sites had populations that were most similar to the reference strain. However, we did not explicitly search for specific SNVs that were enriched in one habitat vs. another. Future studies of nucleotide and codon variants across habitats will reveal the importance of nucleotide- and amino-acid level changes for habitat specialization (Delmont et al., 2019). “

3. One of the major observations of this study is the limited capability of cultivated organisms to represent population in an environment, which is not new.

We agree with the reviewer that this major observation is not new. However, we hope to convey that the increased specificity allowed by combining metagenomes with genomes provides a framework for directly identifying the genes in cultivar genomes that are underrepresented in an environment relative to their core. This new level of resolution offers the possibility for researchers using isolates for laboratory experiments to precisely identify which genes are and aren't represented in natural populations, providing greater transparency between in vitro experiments and in vivo ecology. We have clarified this subtle but important distinction by editing the abstract to read (Lines 16-18):

“... we identified not only limitations in the ability of cultivated organisms to represent populations in their native environment, but also specifically which cultivar gene sequences were absent or ubiquitous.”

and by adding the following paragraph (Lines 624-629) to the discussion:

“A major benefit of our metapangenomic strategy is that it permits us to identify which genes and cultivars are most representative of the microbiota growing in a given natural habitat. We used metapangenomics to query each gene from the *Rothia* and *H. parainfluenzae* cultivar pangenomes across metagenomes from the human oral cavity and measure the abundance of each cultivar gene across environments. These data can serve as a resource to guide the selection of the most environmentally representative strains and gene sequences for future experiments.”

4. Any genomic characteristics of the *Rothia* habitat-specialized species/subspecies/subgroups, such as functional groups? While there are number of gene clusters as unique/core, no discussion on functional aspect was included?

We thank the author for this excellent suggestion that will improve the manuscript. We have added a supplemental table and figure for *Rothia* documenting the gene functions enriched in each species, which is summarized in the main text in Lines 410-411:

“Investigating the predicted functions core to each species also supported the observed differentiation of species (Supplemental Text, Additional File 5C).”

And the specific results and interpretation are provided in the following Supplemental Text paragraph (Lines 1403-1419):

“Functional enrichment analysis indicated that *Rothia* species with similar gene cluster content also contained similar functions. Predicted TIGRFAM functions were used to apply the same functional enrichment analysis as for *H. parainfluenzae*, but this time the groups were the three *Rothia* species (Additional File 5C, Additional File 12). Unlike the *H. parainfluenzae* analysis, the number of genomes per group varied much more substantially, with 48 *R. mucilaginosa*, 15 *R. dentocariosa*, and 4 *R. aeria* genomes. Yet, *R. dentocariosa* and *R. aeria* were still more functionally similar than either were to *R. mucilaginosa* based on aggregate enrichment scores (Additional File 5C), agreeing with the similarity of *R. dentocariosa* and *R. aeria* genomes based on gene cluster content (Figure 3 dendrogram).

The functions enriched in each species also revealed possible sources of niche differentiation. Two functions were found in all 15 *R. dentocariosa* genomes but no other *Rothia* species, a PTS-system sucrose transporter component and a transcription repressor gene (Additional File 12). Further, of the 13 functions core to all *R. dentocariosa* and *R. aeria* genomes but absent from all *R. mucilaginosa* genomes, three were cytochrome related (Additional File 12). As both *R. dentocariosa* and *R. aeria* appear most abundant in plaque (Figure 3 heatmap), these cytochrome differences relative to *R. mucilaginosa* could potentially reflect selection by the different oxygen conditions of their respective microhabitats within tongue and plaque habitats.”